# Effect of FKBP12-Derived Intracellular Peptides on Rapamycin-Induced FKBP–FRB Interaction and Autophagy

**DOI:** 10.3390/cells11030385

**Published:** 2022-01-24

**Authors:** Carolina A. Parada, Ivan Pires de Oliveira, Mayara C. F. Gewehr, João Agostinho Machado-Neto, Keli Lima, Rosangela A. S. Eichler, Lucia R. Lopes, Luiz R. G. Bechara, Julio C. B. Ferreira, William T. Festuccia, Luciano Censoni, Ivarne Luis S. Tersariol, Emer S. Ferro

**Affiliations:** 1Departments of Pharmacology, Biomedical Sciences Institute, University of São Paulo, São Paulo 05508-900, SP, Brazil; casp.biomed@hotmail.com (C.A.P.); ivan.pires.oliveira@gmail.com (I.P.d.O.); ferrari.mayaracalegaro@gmail.com (M.C.F.G.); jamachadoneto@gmail.com (J.A.M.-N.); kelilima@usp.br (K.L.); reichlerusp@gmail.com (R.A.S.E.); llopes@usp.br (L.R.L.); 2Institute of Agricultural Sciences, Federal University of Minas Gerais, Montes Claros 39400-149, MG, Brazil; 3Departments of Anatomy, Biomedical Sciences Institute, University of São Paulo, São Paulo 05508-900, SP, Brazil; luizbechara@yahoo.com.br (L.R.G.B.); jcesarbf@usp.br (J.C.B.F.); 4Departments of Physiology, Biomedical Sciences Institute, University of São Paulo, São Paulo 05508-900, SP, Brazil; william.festuccia@usp.br; 5Department of Physical Chemistry, Institute of Chemistry, University of Campinas, Campinas 13083-970, SP, Brazil; luciano.censoni@gmail.com; 6Department of Biochemistry, Paulista Medical School, São Paulo Federal University (UNIFESP), São Paulo 04021-001, SP, Brazil; ivarne.tersariol@gmail.com

**Keywords:** intracellular peptides, protein–protein interaction, mTORC1, edgotype

## Abstract

Intracellular peptides (InPeps) generated by proteasomes were previously suggested as putative natural regulators of protein–protein interactions (PPI). Here, the main aim was to investigate the intracellular effects of intracellular peptide VFDVELL (VFD7) and related peptides on PPI. The internalization of the peptides was achieved using a C-terminus covalently bound cell-penetrating peptide (cpp; YGRKKRRQRRR). The possible inhibition of PPI was investigated using a NanoBiT^®^ luciferase structural complementation reporter system, with a pair of plasmids vectors each encoding, simultaneously, either FK506-binding protein (FKBP) or FKBP-binding domain (FRB) of mechanistic target of rapamycin complex 1 (mTORC1). The interaction of FKBP–FRB within cells occurs under rapamycin induction. Results shown that rapamycin-induced interaction between FKBP–FRB within human embryonic kidney 293 (HEK293) cells was inhibited by VFD7-cpp (10–500 nM) and FDVELLYGRKKRRQRRR (VFD6-cpp; 1–500 nM); additional VFD7-cpp derivatives were either less or not effective in inhibiting FKBP–FRB interaction induced by rapamycin. Molecular dynamics simulations suggested that selected peptides, such as VFD7-cpp, VFD6-cpp, VFAVELLYGRKKKRRQRRR (VFA7-cpp), and VFEVELLYGRKKKRRQRRR (VFA7-cpp), bind to FKBP and to FRB protein surfaces. However, only VFD7-cpp and VFD6-cpp induced changes on FKBP structure, which could help with understanding their mechanism of PPI inhibition. InPeps extracted from HEK293 cells were found mainly associated with macromolecular components (i.e., proteins and/or nucleic acids), contributing to understanding InPeps’ intracellular proteolytic stability and mechanism of action-inhibiting PPI within cells. In a model of cell death induced by hypoxia-reoxygenation, VFD6-cpp (1 µM) increased the viability of mouse embryonic fibroblasts cells (MEF) expressing mTORC1-regulated autophagy-related gene 5 (Atg5), but not in autophagy-deficient MEF cells lacking the expression of Atg5. These data suggest that VFD6-cpp could have therapeutic applications reducing undesired side effects of rapamycin long-term treatments. In summary, the present report provides further evidence that InPeps have biological significance and could be valuable tools for the rational design of therapeutic molecules targeting intracellular PPI.

## 1. Introduction

Proteins rarely function alone; rather, they interact with general macromolecular components of the cells [1,2]. Edgotype, a fundamental link between genotype and phenotype [3], are regulated by such macromolecular interactions orchestrated by proteins [3]. There are several regulatory mechanisms that make protein–protein interactions (PPI) dynamic and transient, phosphorylation and proteolysis being the ones most widespread [4]. Targeting PPI is a direction in treating diseases and an essential strategy for the development of new drugs [5,6,7,8].

Proteomic pipelines usually utilize enzymatic digestion to generate peptides for the subsequent mass spectrometry analysis of proteins sequences [9]. Peptidomics, on the other side, allows for establishing the signature of naturally occurring endogenous peptides (peptidome) without prior enzymatic digestion [10,11,12]. An essential system that controls protein stability in the nucleus and cytosol is the ubiquitin-proteasome proteolytic system [13,14]. Indeed, evolutionary ancient functions for peptides processed by proteasomes can be evidenced, considering that proteasome-structurally-related proteases are present in lower organisms such as prokaryotes [15,16,17]. A large group of peptides directly identified by mass spectrometry derived from the proteasome degradation of intracellular proteins (i.e., peptides derived from the cytosolic and/or nuclear proteins) were shown to be functional and termed intracellular peptides (InPeps) [7,10,18,19,20,21,22,23,24,25]. InPeps correspond to half of the peptide content of murine brain [26] and are present in several human cell lines [11] as well as in the human brain [27], liquor [12], and brown adipose tissue [28]. InPeps are not part of the cellular degradome and are often found altered in human diseases [12], and a variety of studies has raised the possibility that they may be involved in multiple biological functions, including the regulation of PPI [22,25]. InPeps derived from cytosolic proteins can be secreted [29,30] and were shown to interact with membrane receptors [31,32]. Non-secreted InPeps, corresponding to nearly 80% of the total InPeps identified in cultured mouse brain [29], were suggested to affect cell signaling through the modulation of PPI [18,22,33,34,35,36]. Recently, using structural modeling and computational analysis, InPeps were suggested to interact with microRNAs to regulate gene expression [12,37]. Therefore, InPeps have been extensively shown to have biological and pharmacological significance.

Autophagy was originally defined as the delivery of cytoplasmic cargo to the lysosome for degradation. There are at least three distinct forms of autophagy (i.e., chaperone-mediated autophagy, microautophagy, and macroautophagy), which differ in terms of mode of cargo delivery to the lysosome [38]. Macroautophagy is the major catabolic mechanism used by eukaryotic cells to maintain nutrient homeostasis and organellar quality control. It is mediated by a set of evolutionarily conserved genes, the autophagy-related genes (Atg). With a few exceptions, all Atg are required for the efficient formation of sealed autophagosomes that proceed to fuse with lysosomes [38]. The mechanistic target of rapamycin complex 1 (mTORC1) is a member of the phosphatidylinositol 3-kinase-related kinase family of protein kinases [39]. Under nutrient-rich conditions mTORC1 promotes cell growth by stimulating the synthesis of proteins, lipids, and nucleotides and by inhibiting cellular catabolism through repression of the autophagic pathway [39,40,41]. Consensus is increasing that mTORC1 activation occurs at the surface of the lysosomal membrane in response to changes in amino acid sufficiency, which is transduced via the Rag family of small GTPases to mediate the translocation of mTORC1 from the cytoplasm to the surface of the lysosome, where mTORC1 is activated by GTP-binding protein Rheb [42]. mTORC1 integrates various stimuli and signaling networks, while autophagy constitutes an important avenue for nutrient supply, providing amino acids to restore mTORC1 activity. mTORC1 localization to the lysosome in autophagy is subject to intricate control by metabolic regulatory networks [42,43,44]. The activity of mTORC1 toward many substrates is acutely sensitive to rapamycin, but mTORC1 also possess rapamycin-resistant activity toward certain substrates [45,46,47].

The protein complex formation between 12-kDa FK506-binding protein (FKBP12) and FKBP12-rapamycin binding domain (FRB) results in mTORC1 inhibition by restricting substrate access to the mTORC1 serine/threonine-protein kinase site [48,49]. Rapamycin shows many beneficial effects in mice, while in humans rapamycin and rapamycin derivatives (rapalogs) have been used primarily as immunosuppressants following organ transplantation and in the treatment of several specific types of cancer, including renal cell carcinoma, pancreatic neuroendocrine tumors, and HER2-negative breast cancer [47]. The inhibition of the mTORC1 pathways by rapamycin and rapalogs has also been suggested to extend lifespan [50]. Rapamycin has also clinical applications as a medication to prevent organ transplant rejection and was approved by the FDA to preserve renal allografts under the generic name sirolimus [42,51]. Serious side effects of rapamycin and rapalogs observed in humans include an increased incidence of viral and fungal infections including pneumonia, chronic edema, painful oral aphthous ulceration, and hair loss [52,53]. Metabolic effects of long-term rapamycin treatment have also been observed, including decreased insulin sensitivity, glucose intolerance, and an increased risk of new-onset diabetes [47,53].

The relative levels of InPeps VFDVELLKLE (VFD10) and VFDVELL (VFD7), derived from FKBP12, were largely reduced in HEK293 cells following proteasome inhibition [21,23,24]. VFD10 and VFD7 were also highly elevated in the cortex of Purkinje cells degeneration mice, suggesting a yet non characterized biological significance in neurodegeneration [25]. VFD10 and VFD7 (10–50 µM) were shown by surface plasmon resonance to inhibit PPI between cytosolic brain proteins and both calmodulin (CaM) and 14-3-3 epsilon [35]. VFD7 (5–20 µM) introduced into the HEK293 cells by means of transient membrane permeabilization induced by 3-[(3-cholamidopropyl) dimethylammonio]-1-propanesulfonate (CHAPS) was shown to increase cytosolic Ca^2+^ concentrations [35]. Cell-penetrating peptides (cpp) are relatively short peptides, 4–40 aa, frequently used to carry associated molecules into the cells mainly by endocytosis [54]. The covalent coupling of peptides to cpp constitutes a useful alternative to CHAPS transient membrane permeabilization, allowing easier cell internalization to investigate possible intracellular functions of peptides [55,56,57].

Here, commercially available (NanoLuc Binary Technology, NanoBiT^®^; Promega, WI, USA) structural complementation reporter system, constructed from deep sea shrimp *Oplophorus gracilirostris* small luciferase subunits [58,59], was used to investigate the effect of VFD7/VFD7-cpp and derivatives on the formation of the FKBP–rapamycin–FRB complex that within cells results in mTORC1 inhibition [48]. The potency of rapamycin to induce FKBP and FRB protein dimerization was previously shown to be minimally affected by FKBP–FRB intrinsic affinity (range > 100,000) [58]. The cpp (YGRKKRRQRRR) used herein was previously shown to transport its cargo mainly into the cytosol [60,61,62]. VFD7-cpp (10–500 nM) and FDVELL (VFD6) covalently coupled to cpp (VFD6-cpp; 1–500 nM) were the most potent peptides inhibiting rapamycin-induced FKBP–FRB interaction measured using the NanoBiT^®^ technology. Molecular dynamics (MD) simulations suggested a possible mechanism for VFD7-cpp and VFD6-cpp to inhibit rapamycin-induced FKBP–FRB, which involves FKBP structural change and/or the direct binding of these peptides to the FKBP–FRB protein interaction domains. VFD6-cpp (1 µM) increased cell viability in Atg5^+/+^, but not autophagy-deficient Atg5^−/−^ mouse embryonic fibroblasts. Moreover, extraction of InPeps using different conditions contributed with a possible mechanism for their intracellular proteolytic stability and also corroborate with the suggestions that InPeps can be natural modulators of protein interactions within cells [18,34,63,64].

## 2. Experimental Procedures

### 2.1. Cell Culture

Cell culture was used to investigate the possible effect of peptides (i.e., VFD7-cpp and derivatives) on PPI. Human embryonic kidney 293 (HEK293) cells were chosen because of their reliable growth and propensity for transfection, as well as because these immortalized human embryonic kidney cells have been frequently used for peptidomics and cell biology research for many years [11,21,24,65]. HEK293 cells used herein were purchased from ATCC (Manassas, VA, USA) and were cultured at 37 °C, 5% CO_2_ in high glucose Dulbecco’s Modified Eagle Medium (DMEM; Gibco, Gaithersburg, MD, USA), supplemented with 10 % FBS and penicillin 100 U/mL and streptomycin 100 µg/mL (Gibco, Gaithersburg, MD, USA).

### 2.2. Protein Complementation Assay to Investigate the Effect of Peptides on PPI

The protein complementation assay technology used herein, NanoBiT^®^ (Promega, Madison, WI, USA), was commercially designed to investigate inducible PPI under in vivo cellular conditions, allowing the analyses of distinctive molecules that, introduced within cells, can modify inducible PPI [58]. All experimental procedures were performed according to the manufacturer’s recommendations. Briefly, transient transfections were performed with NanoBiT^®^ PPI Control Pair FKBP–FRB (Promega, Madison, WI, USA; catalog number N2016), containing the FRB-LgBit and FKBP-SmBiT vectors (NanoBiT^®^ PPI Control Pair FKBP, FRB; catalog number N2016, Promega, Madison, WI, USA). Plasmids were co-transfected into HEK293 cells using FuGENE^®^ 6 Transfection Reagent (Promega, Madison, WI, USA), according to the recommendations of the manufacturers. Cells transfected in the absence of DNA were the negative controls of the experiments, whereas the green fluorescent protein expressing pEGFP plasmid (Clontech, Mountain View, CA, USA; 100 ng) was used to evaluate transfections efficiencies; after 30 h of transfection, the pEGFP expression was easily assessed by fluorescent microscopy (data not shown).

HEK293 cells (approximately 10^4^) cultured in a 96-well plate in DMEM high-glucose medium, supplemented with 10% FBS and 1% penicillin/streptomycin, were transiently co-transfected with 200 ng of both FRB-pBiT and FKBP-SmBiT vectors (NanoBiT^®^ PPI Control Pair FKBP, FRB; catalog number N2016, Promega, Madison, WI, USA); this final plasmid concentration of 200 ng was defined previously testing different plasmid concentrations, as shown in Appendix A. The transfected cells were incubated in Opti-MEM^®^ I Reduced Serum Medium culture medium (Gibco, Gaithersburg, MD, USA) for 2 h and then treated with 200 nM of rapamycin (BPSBioscience, CAS 53123-88-9, San Diego, CA, USA) diluted in Opti-MEM^®^/0.01% DMSO (Gibco, Gaithersburg, MD, USA; this rapamycin concentration of 200 nM was defined previously testing different concentrations, as shown in Appendix A). The indicated concentration of peptides was added simultaneous with the rapamycin; after 30 min a booster with the same initial concentration of the indicated peptide was added to the cells, because previous studies have suggested the short half-life of many peptides within cells [36]. After adding the rapamycin and peptides, Nano-Glo Live Cells substrate (Promega, Madison, WI, USA) was added, and the luminescence was measured for 60 min in a Centro LB 960 Microplate luminometer (Berthold Technologies, Bad Wildbad, Germany) and displayed in relative light units (RLU). Graphs of RLU versus time were plotted by GraphPad Prism (GraphPad, San Diego, CA, USA). Control experiments were conducted using VFD10 and VFD7 without cpp covalently bound, in addition to cpp alone (Appendix A).

### 2.3. Peptide Synthesis

Synthetic model peptides (Table 1) were used herein to gain insight on the possible biological function of naturally occurring InPeps as well as to evaluate the possible pharmacological therapeutic application of these peptides. Peptides were custom-made, synthesized by Proteimax Biotechnology LTDA (São Paulo, SP, Brazil) with at least 90% purity (typical chromatograms were provided as Appendix A; Appendix A from peptide synthesis analyses). Peptide synthesis resin was performed on pre-loaded Wang resin (free acid C-terminals). Peptide synthesis used standard Fmoc solid-phase syntheses [66,67], with activating reagent 2-(1H-benzotriazol-1-yl)-1,1,3,3,-tetramethyluronium hexafluorophosphate (HBTU) on an ABI 431 synthesizer. Fmoc deprotection was conducted on the instrument with 20% piperidine for about 10 min. Acetic anhydride acetylation was conducted on the instrument for about 10 min. Cleavage and side-chain deprotection was performed using 10 mL TFA/0.5 mL H_2_O/0.5 mL thioanisole/0.25 mL 1,2-ethanedithiol/0.75 g crystalline phenol for 3 h at room temperature. Most of the salts were removed during ether precipitation of the peptide, but trace amounts may remain. Purification was conducted on a Varian HPLC (Agilent, Santa Clara, CA, USA) with YMC reverse phase C18 columns (Merck, Branchburg, NJ, USA), while HPLC analyses were performed using a Vydac 2.1 mm C18 columns (Merck, Branchburg, NJ, USA). A typical purification run used a 13% acetonitrile/0.075% TFA (buffer B) to 50% acetonitrile/0.075% TFA gradient over 30 min to purify this peptide; buffer A was water/0.1% TFA. Counter-ion was trifluoroacetic acid. Peptide elution was followed by absorbance at 214 nm. The synthesis was typically clean with an average recovery of 85%.

Peptides stock solutions at 100 µM were prepared in autoclaved Milli-Q water and kept at −20 °C. Additional dilutions were prepared immediately before the experiments in 20 mM Tris-HCl buffer, pH 7.5 at 25 °C, for intrinsic fluorescence measurements or DMEM for protein complementation assays that investigated the effect of peptides on PPI. Moreover, the rationale to design additional peptides was to identify a minimal VFD7/VFD7-cpp-derived sequence retaining the ability to induce similar effects. Thus, the VFD7-cpp original sequence was successively shortened from the amino terminal, maintaining the “hot spot” leucine residues from C-terminus. This rational was based on previous descriptions that, similar to protein interactions, peptide binding is mediated by “hot spot” hydrophobic residues that have a high frequency of leucine residues [68,69]. Additional modifications included a single amino acid substitution, of aspartic acid (D; side chain pKa 3.9) substituted to either alanine (A; neutral amino acid) or glutamic acid (E; longer side chain, pK 4.3), and also such amino acid substitutions in addition to a shorter N-terminal. These VFD7/VFD7-cpp-derived peptide sequences were designed to gain further insight on the structural specificity of the peptides investigated herein, which is usually an important issue of peptides having complex and flexible structures comparable to those of small and rigid molecules.

### 2.4. Intrinsic Fluorescence Emission to Evaluate Rapamycin-VFD7-cpp Complex Formation

These experiments were performed to evaluate the possible direct molecular interaction between VFD7-cpp and rapamycin, which could interfere with the results observed on the PPI complementation assay within cells. Variations of the intrinsic fluorescence emission of the peptide VFD7-cpp (5 µM) as a function of the rapamycin concentration (0–20 µM), in a 20 mM Tris-HCl buffer, pH 7.5 at 25 °C, were determined. The peptide was excited at 275 nm, and its emission spectrum was analyzed between 290 and 360 nm in a Shimadzu RF-6000 spectrofluorometer (Shimadzu Corporation, Kyoto, Japan).

### 2.5. Structural Models for the VFD7-cpp and Derivatives

To predict the possible secondary structures of the peptides in solution, structural models of VFD7-cpp and selected derived peptides were conducted using different algorithms. VFD7-cpp and additional selected sequences were run through the IUPred2A server [70] and the DisEMBL server [71] to check for the presence of disordered regions. VFD7-cpp and derivative sequences were also submitted to the PEP-FOLD3 tool of the Mobyle framework [72] to obtain secondary structure predictions as well as structural models for each peptide. Next, these structural models’ predictions were used to perform the molecular dynamics (MD) simulations, as described below.

### 2.6. MD Simulations

MD simulations were performed to gain further insights on the molecular mechanism responsible for the inhibition of rapamycin-induced FKBP–FRB interaction, evaluated in HEK293 cells using the NanoBiT^®^ protein complementation assays; these latter assays investigated in vivo the effect of peptides on PPI. The crystallographic structure of the FKBP–rapamycin–FRB complex was obtained from the Protein Data Bank (PDB) database, with a resolution of 1.67 Å (PDB code 5GPG) [73]. Subsequently, to study the impact of the peptides on the FKBP–rapamycin–FRB complex, we kept (i) both proteins complexed with rapamycin (using the x, y, and z coordinates of 5GPG structure) in the presence of forty peptide molecules (VFD7-cpp) in aqueous solution (24,650 water molecules) in a cubic box of side 98 Å. Conversely, to evaluate the disturbance promoted by peptides in the complex formation, the ligand rapamycin was removed from the initial structure 5GPG. Then, FKBP and FRB were separated by approximately 40 Å from each center of mass, adding thirty peptide molecules, seventy-five rapamycin molecules, and 52,500 water molecules into a cubic box of side 123 Å for (ii) VFD7-cpp, (iii) VFD6-cpp, (iv) VFA7-cpp, and (v) VFE7-cpp. In all systems, chlorine and sodium ions were added to neutralize any residual charges. All of the peptides, water molecules, and ions were randomly introduced to obtain a density close to 1.0 g/mL. The initial configurations were built with Packmol [74], containing the proteins FKBP and FRB centered in the simulation box. Subsequently, the trajectories were calculated as follows: (i) keeping all atoms of the proteins fixed, water molecules, rapamycin, peptides, and ions were relaxed performing 10,000 minimization steps by the Gradient-Conjugate (GC) method followed by 200 ps of MD simulation; (ii) keeping only the Cα atoms of the proteins FKBP and FRB fixed, 10,000 GC minimization steps were performed, followed by another 200 ps of MD simulation; (iii) all protein atoms were released and 10,000 GC steps were conducted, with subsequently 20 ps of MD simulation; (iv) an equilibration was conducted before the production run by executing 20,000 steps of GC energy minimization, followed by 0.5 ns of MD simulation; and (v) finally, with the coordinates and final velocities obtained from the equilibration step, independent production runs of 200 ns were computed, totalizing 1 µs of MD simulation. The trajectories were collected in an NPT ensemble at 1 atm and 310.15 K, with the pressure controlled by a Nosé–Hoover–Langevin bath with a damping coefficient of 10 ps^−1^, adopting a timestep of integration of 2 ps. The CHARMM force field was used for proteins FKBP and FRB, peptides, rapamycin, ions, and water molecules [75,76,77,78] using the TIP3 model for water [79]. The molecular parameters for rapamycin were obtained from homology according to the CGenFF platform [80,81]. The trajectories were computed using the NAMD program [82] and all the results, including visualizations, were calculated using MDAnalysis [83] and/or VMD [84].

### 2.7. InPeps Extraction and Quantification

These experiments were intended to assess the fraction of naturally occurring InPeps associated to macromolecular components of the cells (i.e., proteins and/or nucleic acids), which could contribute both to their proteolytic stability and biological function; thus, it is conceivable that peptides associated to proteins can regulate PPI and, at the same time, be protected from proteases and peptidases that hydrolyze peptides/proteins as single-chain. To investigate the possible presence of InPeps associated to cellular macromolecular components, different experimental conditions were used to extract peptides from HEK293 cells (Figure 1). HEK293 subconfluently cells were grown as described above, collected using a cell scrapper, and pelleted by centrifugation. The InPeps extraction protocol follows the scheme described (Figure 1). Peptide concentration in the peptide extracts described above was determined at pH 6.8 using fluorescamine, as previously described [36,85]. The reaction was performed at pH 6.8 to ensure that only the amino groups of peptides and not those of free amino acids reacted with fluorescamine [85]. Briefly, 2.5 μL of sample was mixed with 25 μL of 0.2 M phosphate buffer (pH 6.8) and 12.5 μL of a 0.3 mg/mL acetone fluorescamine solution. After vortexing for 1 min, 110 μL of water was added, and fluorescence was measured with a SpectraMax M2e plate reader (Molecular Devices, San Jose, CA, USA) at an excitation wavelength of 370 nm and an emission wavelength of 480 nm. A peptide mixture of known composition and concentration (containing equimolar concentration of peptides LTLRTKL, DITADDEPLT, SPQLEDEAKEL, and HDSFLKAVPSQKRT) was used as the standard reference for determining the peptide concentration in HEK293 extracts, as previously described [36,86].

### 2.8. Measurement of Cellular Viability after Hypoxia-Reoxygenation in Cell Culture

These experiments were performed to evaluate the possible pharmacological application of VFD7-cpp and additional selected peptides on cell viability, following an ischemic challenge. These experiments also contributed to investigate the possible biological significance of InPeps for cell viability following hypoxia-reoxygenation. Therefore, herein the effect of peptides was evaluated in wild type mouse embryonic fibroblasts cells (MEF) expressing Atg5 (Atg5^+/+^) and in the MEF missing autophagy pathway due to the lack of Atg5 (Atg5^−/−^); Atg5 is indispensable in both canonical and non-canonical autophagy [87,88]. Atg5^+/+^ and Atg5^−/−^ MEF were cultured in a 96-well plate in DMEM high-glucose medium supplemented with 10% FBS and 1% penicillin/streptomycin, as previously described [88]. Cells were washed with PBS and incubated at 37 °C for 3 h in hypoxia buffer (NaCl 125 mmol/L, KCl 8 mmol/L, KH_2_PO_4_ 1.2 mmol/L, MgSO_4_ 1.25 mmol/L, CaCl_2_ 1.2 mmol/L, NaHCO_3_ 6.25 mmol/L, sodium lactate 5 mmol/L, and Hepes 20 mmol/L, pH 6.6) in a GasPak pouch (Becton Dickinson). Then the cells were re-exposed to oxygen, and the buffer was replaced by reoxygenation Krebs–Henseleit buffer (NaCl 110 mmol/L, KCl 4.7 mmol/L, KH_2_PO_4_ 1.2 mmol/L, MgSO_4_ 1.25 mmol/L, CaCl_2_ 1.2 mmol/L, NaHCO_3_ 25 mmol/L, glucose 15 mmol/L, and Hepes 20 mmol/L, pH 7.4) for 3 h. Treatments with 1μM of each peptide (cpp, VFD6-cpp, VFD7-cpp, VFA7-cpp, or VFE7-cpp) or control cpp alone commenced 30 min before ischemia and continued during hypoxia and reoxygenation. A booster with the same initial concentration of the peptide was added with the hypoxia buffer incubation and the reoxygenation buffer incubation. After the protocol, cell viability was measured by using tetrazolium salt 3-(4,5-Dimethylthiazol 2-yl)-2,5-diphenyltetrazolium bromide (MTT) assay; these assays measure the mitochondrial metabolic rate and indirectly reflect the viable cell numbers, which are the most commonly used method to evaluate the effect of compounds on cell viability on cultured cells [89]. MTT-based assay (Cell Counting Kit-8, Dojindo; Merck, São Paulo, SP, Brazil) was conducted according to the manufacturer’s instructions.

### 2.9. Statistical Analysis

Statistical analyses were performed to evaluate possible differences between different experimental groups. Data are presented as mean ± standard error of the mean (SEM). GraphPad Prism 8 software (GraphPad Inc., San Diego, CA, USA) was used to conduct the appropriate unpaired t-test or ANOVA one-way analysis of variance followed by Tukey’s post-hoc test to identify possible differences between groups. A value of *p* < 0.05 was considered statistically significant.

## 3. Results

### 3.1. Intrinsic Fluorescence Emission Analyses of the Direct Interaction of VFD7-cpp and Rapamycin

VFD7-cpp was investigated in order to evaluate its possible molecular interaction with rapamycin. Such direct molecular interaction could interfere with results obtained on the cellular complementation assays evaluating FKBP–FRB interaction induced by rapamycin within HEK293 cells. VFD7-cpp was excited at 275 nm, and its emission spectrum was analyzed, as a function of rapamycin concentration. Results suggested that rapamycin promotes the suppression of the fluorescence emission of Phe (301 nm), as well as Tyr (314 nm), from VFD7-cpp; the fluorescence variation was analyzed both at 301 nm and at 314 nm as a function of the rapamycin concentration (Figure 2). Peptide fluorescence variations both at 301 nm and 314 nm as a function of rapamycin concentration were saturable and obeyed the equilateral hyperbole equation (Michaelis–Menten type), providing a similar dissociation constant (Kd). The Kd analyzed by the change in Phe fluorescence (301 nm) was 11.4 ± 0.9 µM, and the Kd analyzed by the change in Tyr fluorescence (314 nm) was 14.5 ± 1.3 µM. An apparent *K*_d_ of 13 ± 1 µM was determined by VFD7-cpp-rapamycin interaction. These values suggest that rapamycin can interact with the VFD7-cpp peptide at a single site, with saturable and reversible binding (Figure 2). To avoid the possible interference of VFD7-cpp and rapamycin direct interaction, further experiments using the NanoBiT^®^ structural complementation reporter system were conducted with peptides at concentrations 200-fold lower (i.e., 1–500 nM) than VFD7-cpp/rapamycin apparent Kd’s of 13 ± 1 µM.

### 3.2. Effect of VFD7-cpp and Derivatives on Rapamycin-Induced FKBP–FRB Interaction

Next, to evaluate the effect of VFD7-cpp and derived peptides on PPI, HEK293 cells were transiently transfected with NanoBiT^®^ plasmid vectors encoding either FKBP or FRB [58], and the FKBP–FRB interaction was induced using rapamycin; different concentrations of the plasmids vectors and rapamycin were previously tested to determine appropriate concentrations to perform further experiments (Appendix A). Neither VFD7 nor VFD10 without the cpp covalently bound, nor the cpp alone, were able to inhibit the rapamycin-induced FKBP–FRB interaction (Appendix A). These data suggesting that without cell internalization, VFD7 and derived peptides, as well as cpp alone, have no effect inhibiting rapamycin-induced FKBP–FRB interaction within HEK293 cells.

The design of peptides for therapeutic application involves at least a rational step to identify derivatives with increased potencies and a smaller number of amino acids; these modifications could reduce the later costs of using peptides in pharmaceutical formulations. Here, the rationale of maintaining C-terminus “hot spot” leucine residues [68,69] was partially efficient. Next, the possible effects of VFD7-cpp and derived peptides covalently coupled to cpp on the rapamycin-induced FKBP–FRB interaction were qualitatively evaluated, measuring the RLU generated as a result from FKBP–FRB intracellular interaction (Figure 1A–D). Among peptides evaluated, VFD6-cpp (1–500 nM) followed by VFD7-cpp (10–500 nM) were the most potent inhibiting FKBP–FRB interaction (Figure 2A–D). Peptides VFD5-cpp and VFD4-cpp, although with reduced potencies compared to VFD6-cpp or VFD7-cpp, could inhibit rapamycin-induced FKBP–FRB interaction (Figure 3). However, the presence of these leucine “hot spots” needed to be allied with the presence of the aspartic acid, as the peptides AVE5-cpp and EVE5-cpp, derived from VFD5-cpp, have completely abolished their inhibition of rapamycin-induced FKBP–FRB interaction (Figure 3). Therefore, VFD6-cpp could be suggested as the most potent and smaller VFD7-cpp derivative, capable to inhibit the rapamycin-induced FKBP–FRB interaction.

### 3.3. Structural Models for the VFD7-cpp and Derivatives

Using the IUPred2A server [70], long disordered regions were predicted to exist between residues 1 to 5 and 13 to 18 of VFD7-cpp, VFA7-cpp, and VFE7-cpp, while VFD6-cpp peptide seem potentially disordered throughout (Appendix A). Peptide sequences submitted herein to disordered region predictions by the DisEMBL server [71] produced predictions that were in agreement with those obtained from the IUPred2A server (Appendix A). Secondary structure predictions using the PEP-FOLD3 tool of the Mobyle framework [72] suggested a probable helical conformation for the common cpp at the C-terminal regions of the peptides, while the varying N-terminal regions shows a corresponding greater variability for the first five residues and a probable coil conformation for the (L)LYG fragment (Appendix A).

Despite the relative variability in the secondary structure predictions, helical configurations were overrepresented in the model ensembles generated for the peptides analyzed (VFD7-cpp, VFD6-cpp, VFA7-cpp, and VFE7-cpp). We selected two distinct models for each of these latter peptides in order to represent regions of larger variability while accounting for the secondary structure predictions (Appendix A). All four peptides exhibited a preference for helical conformations at the common cpp sequence from the C-terminal region (Appendix A). For the N-terminal region, the VFA7-cpp peptide adopts mostly a helical conformation, while the VFD7-cpp and VFD6-cpp peptides both exhibited an extended conformation for the first five residues followed by a loop into the helical C-terminal region. The VFD6-cpp peptide shows a relatively greater flexibility, which is consistent with the predicted larger probability for it being disordered. The VFE7-cpp peptide appears to show the greatest preference for extended conformations (Appendix A).

### 3.4. Molecular Dynamics to Evaluate the Effect of the VFD7-cpp and Related Peptides on Rapamycin-Induced FKBP and FRB Interaction

To gain further insight about the structural relationship of VFD7-cpp and VFD6-cpp inhibiting with greatest potencies the rapamycin-induced FKBP–FRB interaction, MD simulations were performed to obtain a detailed picture of the interactions between these peptides with both the FKBP and FRB proteins, as well as to identify possible disturbances promoted by the peptides to the FKBP–rapamycin–FRB complex. Peptides VFA7-cpp and VFE7-cpp of similar secondary structures and amino acid sequences, although producing no inhibition of rapamycin induced FKBP–FRB interaction, were used as “specificity control” peptides.

Simulations performed with pre-stablished FKBP–rapamycin–FRB complex suggested that VFD7-cpp was not able to cause dissociation of this complex (Figure 4A); note that VFD7-cpp, green line, did not disturb these interactions, keeping the distance close to 20 Å from each protein center of mass. On the other hand, VFD7-cpp prevents the formation of the FKBP–rapamycin–FRB complex in a solution of peptides and rapamycin (Figure 4B, green line on the upper right panel). These data corroborate previous experimental results obtained in vitro using surface plasmon resonance, showing that VFD7 inhibits the interaction but does not disrupt the association between thimet oligopeptidase and CaM [35]. VFD7-cpp kept the FKBP and FRB proteins separated by approximately 40 Å during almost all the simulation time (Figure 4B, upper middle panel). The other peptides (VFD6-cpp, VFA7-cpp, and VFE7-cpp) were also able to maintain this distance between FKBP and FRB, although not as stable as for VFD7-cpp; that was particularly evident for the VFE7-cpp peptide. These findings suggest a possible disturbance in protein conformation induced by VFD7-cpp, which could contribute to the observed inhibition of rapamycin-induced FKBP–FRB interaction within HEK293 cells.

Corroborating the results above, the previously formed FKBP–rapamycin–FRB complex was not disturbed by the VFD7-cpp peptide, as indicated by the displacement analysis root-mean-square deviation (RMSD; Figure 5A), although, a slight conformational change to stabilize the system was observed at around 100 ns (see green line). However, when the FKBP and FRB proteins were separated by 40 Å, the presence of the VFD7-cpp peptide upheld the separation of the proteins beyond their initial coordinates (Figure 5A, see the black line). These data suggest that VFD7-cpp interfere with the well-oriented conformations of both proteins, hindering the formation of the complex. Next, the possibility of induced rotation was detailed for both FKBP and FRB proteins considering each peptide (Figure 5B–E). The VFD7-cpp peptide seem to alter substantially the initial conformation of FKBP, with comparatively less disturbance promoted on the FRB (Figure 5B). The rotations induced by additional peptides VFD6-cpp, VFA7-cpp, and VFE7-cpp seem less significant (Figure 5C–E) even though a small rotation could be induced by VFD6-cpp peptide on FKBP, as seen around 175–200 ns (Figure 5C).

These data suggest that VFD7-cpp and VFD6-cpp could be preventing the FKBP–FRB protein–protein interaction complex induced by rapamycin, altering the structure of FKBP–FRB needed for appropriated orientation in protein–protein complex formation; no important changes in such orientation were observed for peptides VFA7-cpp or VFE7-cpp. A functional consequence of such structural disturbance could be the absence of the heterodimer FKBP–FRB formation, leading to mTORC1 signaling modulation [48,90].

### 3.5. Impact on the FKBP Conformation Induced by the VFD7-cpp Peptide

The FKBP rotation induced by VFD7-cpp (Figure 5B) can be understood by the binding of this peptide to the FKBP–FRB complex interface (Figure 6). Clearly, the VFD7-cpp peptide influences the well-oriented PPI interfaces of FKBP–FRB (Figure 6A, arrows). From this point, it is plausible to suggest that the perpendicular arrangement may disfavor the rapamycin mechanism of binding, and this disorientation must decrease the FKBP–FRB complex formation. Rapamycin should interact first with FKBP and afterward with FRB to inhibit the mTORC1 pathway [16]. Here, VFD7-cpp was observed to inhibit the rapamycin-induced FKBP–FRB interaction within cells (Figure 3). The ability of the VFD7-cpp to be anchored close to FKBP interface can be explained by hydrogen interactions between the arginine and aspartate residues present in the VFD7-cpp amino acid chain; the interaction of the terminal arginine of the peptide with a carbonyl group from the Lys170 of the FKBP backbone and the interaction of the hydroxyl side chain of Tyr198 with aspartate (Figure 6B). A similar anchoring mechanism could be occurring for the other peptides investigated herein. Important interactions were observed with the side chain of arginine (from cpp) and the FKBP residues Asp146, Ile149, and Gln150; the peptide always acts as the donor in the interaction (Figure 6B). Recently, Song-Yi Lee and collaborators [4] described the main residues for the FKBP–FRB interactions induced by rapamycin. Residues Asp146-Lys170-Tyr198 make up the interface of binding of the FKBP–FRB domain, suggesting that VFD7-cpp can inhibit the rapamycin-induced interaction of FKBP–FRB through a competitive mechanism binding to this region of FKBP protein surface. Therefore, the peptide may compromise FKBP–FRB dimer formation by keeping the protein structures apart and promoting FKBP rotation that hinders the formation of the FKBP–rapamycin–FRB complex.

### 3.6. Surface Interactions between FKBP and FRB and the Effects of VFD7-cpp and Derived Peptides

A direct approach to understanding the effects promoted by VFD7-cpp, VF67-cpp, VFA7-cpp, and VFE7-cpp could be to evaluate the direct protein–peptide interactions. In this sense, possible VFD7-cpp, VF67-cpp, VFA7-cpp, and VFE7-cpp interactions on FKBP–FRB protein surfaces were investigated and expressed as the average number of peptide molecules close to the proteins (Figure 7A). These data suggest that the number of accumulated peptides binding to the individual or complexed proteins do not match the inhibitory effects observed in HEK293 cells (i.e., VFA7-cpp shows the larger number of peptides binding per protein). Thus, binding to a specific domain (i.e., involved on the rapamycin-induced FKBP–FRB interaction) seem more relevant than the number of binding sites. Despite the frequent interactions observed for all peptides, mainly on the FKBP domain (Figure 7A), the previous RMSD analysis of proteins’ conformational changes show an important rotation of the FKBP structure promoted by VFD7-cpp (Figure 5B) and also a minor one for VFD6-cpp at the end of the MD trajectory (Figure 5C). Moreover, direct protein–peptide interactions suggested that VFD6-cpp binds at the FKBP–FRB interaction domain (Figure 7C). These data could corroborate the greatest potency observed for VFD6-cpp inhibiting rapamycin-induced FKBP–FRB interaction within HEK293 cells. The cpp segment used for the internalization of the peptides into HEK293 cells was frequently observed in a well-oriented conformation to the protein surfaces, especially on the FKBP–FRB interface (Figure 7B–E). This behavior can be understood from the hydrophobicity of both the FKBP–FRB interfaces [4,16] and the cpp complementary portion of the peptides containing the non-polar residues valine, phenylalanine, and leucine. Taken altogether, MD simulations suggest a steric mechanism of inhibition caused by the peptides binding to FKBP and FRB interaction domains. Consequently, in the presence of peptides such as VFD7-cpp or VFD6-cpp, rapamycin could fully induce FKBP–FRB complex formation, making it less effective in inhibiting mTORC1 kinase activity.

### 3.7. InPeps Exist Associated as well as Unassociated/Free from Macromolecular Components of the Cell

In agreement with MD simulations shown above, previous reports suggest the association of InPeps to macromolecular components of the cells, which could contribute to understanding both their intracellular proteolytic stability and biological significance inhibiting PPI within living cells [36,91,92]. Next, to gain further insights into the proteolytic stability and biological significance of InPeps by possibly interacting with macromolecular components of the cells (i.e., proteins and/or nucleic acids), peptide extractions from HEK293 total lysates, slightly modified from previous descriptions [30,36,86,93], were conducted in two rounds. In the first round, peptide extraction from HEK293 cell homogenates was evaluated in the flow-through of the 10,000 Da size-exclusion membranes. During the second round, peptide extraction from HEK293 cell homogenates was evaluated on the material retained on the 10,000 Da size-exclusion membranes (schematized in Figure 1). During the first round of extractions, highest recoveries of peptides obtained in the flow-through were from conditions that denature protein structure (i.e., 80 °C, 8M urea at room temperature, and 8M urea at 80 °C) compared to extractions conducted with PBS at 4 °C (4 °C; smoothest extraction condition), methanol (MeOH), or acetonitrile (ACN) at room temperature (Figure 8). Conversely, the opposite was observed during the second round of extraction, which used the material retained in the 10,000 Da size-exclusion membranes during the first round of extraction. Thus, the highest recovery of peptides from the 10,000 Da size-exclusion membranes was obtained from those membranes used to filter the peptides extracted with PBS at 4 °C (Figure 8). Similar results were obtained extracting peptides from mice brain, spleen, or liver tissues (data not shown). These results support a model that within cells InPeps exist mainly associated to macromolecular components of the cells. Free/unassociated peptides were also extracted using non-denaturing conditions (PBS at 4 °C), suggesting the existence of a homeostatic equilibrium that could also contribute to the presence of InPeps within cells.

### 3.8. Possible Pharmacological Effects of VFD7-cpp and Derived Peptides on Autophagy

Peptides VFD7-cpp and VFD6-cpp were the most potent inhibitors of rapamycin-induced FKBP–FRB interaction within HEK293 cells, while peptides VFA7-cpp and VFE7-cpp have no such inhibitory effect (Figure 3). Therefore, a possible pharmacological application of VFD7-cpp and VFD6-cpp could be to prevent rapamycin-induced FKBP binding to mTORC1 regulatory domain FRB, which could stimulate mTORC1 activity, promoting the relevance of inhibiting eased cell viability [48]; the maintenance of cell viability is one of the functions directly related to mTORC1 [94,95]. Autophagy gene 5 (Atg5) plays a key function in autophagic vacuole formation, being required for autophagic cell death [96,97]; Atg5 interacts with the death domain of Fas-associated protein as a necessary step leading to autophagy cell death [98]. Moreover, previous data suggest that lysosome activation induced by mTORC1 inhibition depends on Atg5 [99].

The functional effects of VFD7-cpp, VFD6-cpp, VFA7-cpp, VFE7-cpp, and cpp (1 µM), were evaluated using MTT assays in MEF cells either expressing Atg5 (Atg5^+/+^) or deficient of Atg5 expression (Atg5^−/−^; autophagy-deficient MEF cell; Figure 8). In addition to cpp as a control peptide, VFA7-cpp and VFE7-cpp that were not effective inhibiting an FKBP–FRB interaction within HEK293 cells (Figure 3) were also used to further evaluate the specificity of the peptides VFD7-cpp and VFD6-cpp on cell viability assays. VFD6-cpp, the peptide with the greatest potency inhibiting FKBP–FRB interaction, was shown to increase cell viability in Atg5^+/+^, but not on autophagy-deficient Atg5^−/−^ MEF cells challenged by hypoxia (Figure 9). The specificity of VFD6-cpp increasing cell viability only on Atg5^+/+^ could be related to its ability to inhibit the FKBP–FRB interaction that could result in mTORC1 activation [99]. However, further experiments should be conducted to investigate this possibility. These assays used peptides of similar structures to find that only VFD6-cpp was capable to improve cell viability, suggesting the large specificity of peptides in producing pharmacological effects.

## 4. Discussion

The data presented here provide novel proof-of-concept that corroborate InPeps’ biological and pharmacological relevance as regulators of PPI. Among the peptides evaluated, VFD7-cpp and VFD6-cpp inhibited rapamycin-induced FKBP–FRB protein interaction with the greatest potencies. VFD6-cpp shows structural specificity increasing Atg5^+/+^ MEF cells viability following hypoxia challenge. These data suggesting the potential application of VFD6-cpp to prevent cell damage follow an ischemic event, as observed in the stroke [100,101]. VFD7-cpp and VFD6-cpp could also be of therapeutic relevance to reduce the undesired side-effects of rapamycin and rapalogs [102,103]. Moreover, InPeps were suggested to exist associated as well as unassociated/free to macromolecular components of the cells. These data could suggest the existence of a homeostatic equilibrium regulating InPeps half-life, which contributes both to natural InPeps proteolytic stability and biological significance. Together, the results presented here suggest that InPeps can be used as prototypes to design novel molecules to target intracellular PPI-based pharmacological therapeutics.

The PPI investigated herein used NanoBiT^®^ technology, designed for the qualitative investigation of protein interaction dynamics under relevant physiological conditions. Benefiting from the small size and bright luminescence of deep-sea shrimp *Oplophorus gracilirostris* small luciferase subunits, it provides PPI detection at low intracellular concentrations with minimal steric interference on appended target proteins [58,59]. The intrinsic affinity and association constants determined for NanoBiT^®^ were described to be outside of the ranges typical for protein interactions [58,59]. For the specific interaction of FKBP–FRB, it showed second-order behavior consistent with a dynamic binary system while at higher concentrations favoring full association of the dimeric complex [58,59]. The luminescent response observed was rapid and reversible without evident lag, indicating that NanoBiT^®^ can react nearly instantaneously to changing intracellular conditions [58,59]. Furthermore, dynamic processes can be monitored for more than one hour due to the stable signal duration of the luminescent reaction [58]. VFD6-cpp (1–500 nM) and VFD7-cpp (10–500 nM) were the most potent peptides inhibiting rapamycin-induced FKBP–FRB dynamic interaction. The inhibition of PPI at nanomolar concentrations corroborates previous suggestions that natural InPeps have biological significance regulating intracellular protein interactions [18,35,36]. At higher doses, VFD5-cpp (500 nM) and VFD4-cpp (500 nM) were also able to inhibit rapamycin-induced FKBP–FRB interactions. Peptides VFD10-cpp, VFE7-cpp, VFA7-cpp, EVE5-cpp, and AVE5-cpp were unable to alter rapamycin-induced FKBP–FRB interactions. The VFD7-cpp amino acid substitution from D to A (VFA7-cpp) or to E (VFE7-cpp) completely abolished its inhibition of FKBP–FRB interactions induced by rapamycin. London and colleagues created a unique nonredundant database of high-resolution structures of peptide–protein complexes (termed peptiDB) to investigate the structural basis of free peptide–protein interactions [69]. Free peptides tend to bind to proteins in a more planar fashion than structured and unstructured proteins do, which may allow peptides to overcome their own configurational entropy loss upon binding [69]. In some cases, protein structure is known to be unstable and could be stabilized only upon binding to the peptides [69]. Similar to protein interactions, peptide binding is mediated by “hot spot” hydrophobic residues that have a high frequency of leucine residues [68,69]. Peptide domains within protein structures have been shown to exist throughout entire protein structures [4,104] and could be released intact after the proteolytic degradation of proteins by proteasome (Ferro, E.S. and Klitzke, C.F., unpublished data). Although the participation of proteasome-generated peptides in PPI could seem obvious [18], most previous studies have not considered the fact that free natural InPeps could affect protein interactions within cells. Coincidently, the most frequently identified InPeps generated by the proteasome, which includes the peptide VFD7, have a hydrophobic amino acid, frequently leucine, at the C-terminus [11,24]. Indeed, substitutions of leucine to alanine totally abolished the cell death activity of peptide WELVVLGKL (pep5) [62]. Herein, the presence of the leucine “hot spot” was considered during the rational design of VFD7-cpp derivatives. Results suggest that preserving the leucine “hot spots” was not enough to preserve peptides derived from VFD7-cpp pharmacological activities as inhibitors of rapamycin-induced FKBP–FRB interactions. The absence of the aspartic acid, as in peptides AVE5-cpp and EVE5-cpp, derived from VFD5-cpp, completely abolished the inhibition of rapamycin-induced FKBP–FRB interaction. Therefore, VFD6-cpp could be suggested as the most potent and smaller VFD7-cpp derivative, capable to inhibit the rapamycin-induced FKBP–FRB interactions.

MD simulations suggested that VFD7-cpp and VFD6-cpp directly interacted with both FKBP and FRB. Upon VFD7-cpp or VFD6-cpp binding, FKBP structures rotate, mispositioning their FRB-interaction domain, which could be related to their inhibition of rapamycin-induced interactions of FKBP–FRB in HEK293 cells. MD simulations also suggested that peptides VFA7-cpp and VFE7-cpp bind to distinctive regions of both FKBP and FRB, whereas experimental data suggested that these predicted interactions could not have disturbed proteins structures enough to be functional. Previously, affinity chromatograph columns were constructed, suggesting that InPeps FE2 and FE3 covalently bound through its NH2 terminus interacted, respectively, with 10 and 6 proteins of related biological function; both FE2 and FE3 were InPeps intracellularly active that modulated isoproterenol and angiotensin II signal transduction [36]. Most of the proteins associated to FE2 and FE3 have functions related to vesicular transport and signal transduction regulation. These data suggesting that binding to multiple protein regions/proteins could be an intrinsic property of peptides. Once peptides bind to protein motifs/domains of related functions, from the same or distinctive proteins, they could act synergically, producing functional consequences of biological/pharmacological relevance [4,104]. Thus, the flexible and complex structure of peptides, compared to that of small molecules of rigid and more specific binding sites, could became an advantage for therapeutic application of peptides as inhibitors of intracellular PPI.

Hundreds of InPeps have been described from the most diverse sources, including plants [105,106,107,108], yeast [10], fish [93], and mammals [7,12,21,22]. Despite of the strong evidence that InPeps are natural constituents of the cells, the expected short half-life of peptides within cells full of peptidases contests their stability and biological significance [91,109,110,111]. Indeed, most synthetic peptides introduced within cells were shown to have a short half-life [36,91]. For example, after 30 min of internalization into Chinese hamster ovarian cells, the concentration of a group of four InPeps was reduced from 1.6% (SSGAHGEEGSARIWKA) to 26% (GSAKVAFSAIRSTNH), whereas a random peptide (VNMVPVGWASFR) used as the control was 100% recovered [36]. On the other hand, synthetic fluorescent peptides were shown to compete with endogenous peptides in the nuclear compartment for binding to chromatin [91], which decreases the rate of degradation and suggests the association of InPeps to macromolecular components of the cells [36,91,92]. This paradox has been limiting a clear understanding about the possible biological significance of InPeps. Herein, the extraction of InPeps using different conditions such as heat and 8M urea, well-known to denature proteins, were shown to be more efficient compared to mild extraction with PBS at 4 °C. On the other hand, a significant higher recovery of peptides occurred from the 10,000 Da size-exclusion filter membranes if the peptide extraction was conducted at mild conditions using PBS at 4 °C. These data suggest that within cells, most InPeps were tightly associated to macromolecular components (i.e., proteins and/or nucleic acids), justifying the need for stringent conditions (i.e., heat, 8M urea, and acidic pH) to extract these peptides from cells or tissues [10,11,25,29,112]. These data corroborate previous suggestions that within cells InPeps can scape post-proteasomal proteolysis associating with macromolecular components [91,92]. In addition, these results suggested that within cells InPeps were also unassociated/free from macromolecular components of the cells (i.e., peptides extracted in PBS at 4 °C). Thus, one possibility to explain the presence of natural InPeps within cells and tissues, in opposition to synthetic peptides introduced into the cells that last for only a short time, could be their binding to macromolecular components of the cells. In addition, to maintain the levels of natural InPeps within cells under strict regulatory control, a homeostatic equilibrium must exist [36,65,91,92]. Such mechanisms could help to explain the presence of InPeps within cells and tissues, and the changes observed on InPeps levels in pathological conditions [12,19,20,25,27,113,114], along the different phases of the cell cycle [62] after treatment with interferon gamma [115], induced by thermogenesis in human brown adipose tissue [28], following plant treatment with either phytohormone [105] or salycilic acid [106], during diet-induced obesity [37], among others [65,91,116,117,118,119]. Together, these data corroborate the main suggestion of the present report (using VFD7-cpp and derivatives as a model), that InPeps can be functional within cells binding and regulating PPI.

Autophagy plays a crucial role in maintaining cellular homeostasis and is closely related to the occurrence of variety of human diseases. It is known that autophagy occurs in response to various environmental stresses such as nutrient deficiency, growth factor deficiency, and hypoxia [38]. Many studies have shown that a number of signal transduction pathways are involved in the regulation of autophagy [120,121]. Some autophagic pathways converge at the mTORC1, which plays a central role in the regulation of autophagy [122,123,124,125]. mTORC1 signaling is switched on by several oncogenic signaling pathways and is accordingly hyperactive in the majority of cancers [126], age-related diseases [127], and recently, COVID-19 [128]. Moreover, lysosome activation induced by mTORC1 inhibition depends on Atg5 during autophagy [99]. Rapamycin induces the formation of the FKBP–rapamycin–FRB complex, resulting in the inhibition of some of the effects of mTORC1 [41] by restricting substrate access to the mTORC1 serine/threonine-protein kinase site [48]. VFD6-cpp and VFD7-cpp should be stimulating mTORC1 activity, as the most potent peptides investigated herein to inhibit the rapamycin-induced FKBP–FRB interaction. However, only VFD6-cpp was seen to increase cell viability in Atg5^+/+^, but not on autophagy-deficient Atg5^−/−^ challenged by hypoxia. Inhibition of the mTORC1 with rapamycin is currently the only known pharmacological treatment that increases lifespan in all model organisms studied [129,130,131]. However, evident side effects have been experienced by the chronic use of rapamycin, such as metabolic defects that include hyperglycemia, hyperlipidemia, insulin resistance, and increased incidence of new-onset type 2 diabetes [103]. Patients taking rapamycin to prevent organ transplant rejection have presented with adverse effects including stomatitis, thrombocytopenia, high serum triglycerides and cholesterol, and impaired wound healing [102]. Therefore, a possible therapeutic application for VFD6-cpp could be the inhibition of rapamycin-induced FKBP–FRB interaction and consequent inhibition of mTORC1. This could be useful to reduce at least some of the undesired side-effects of rapamycin and rapalogs. Moreover, VFD6-cpp could be therapeutically useful to prevent extensive cell death during ischemic situations, such as that observed during stroke [100,132]. However, further investigations remain necessary to evaluate whether in living cells proteasome generated InPeps such as VFD7 could be functional stimulating mTORC1 activity.

There are limitations of using VFD6-cpp and VFD7-cpp as therapeutic peptides, including the stability needed to reach intact the intracellular milieu after being administrated and absorbed in vivo. Indeed, it is well known that most pharmaceutical industries and laboratories prefer proteolytic stable small molecules of the well-defined and rigid chemical structure of improved pharmacokinetics. However, it seem relevant to mention that in the past 5–6 years the U.S. Food and Drug Administration (FDA) has authorized the clinical prescription of more than 15 new peptides or peptide-containing molecules [133,134]. Moreover, PPI have been challenging targets for traditional drug modalities, while macrocyclic peptides have proven highly effective PPI inhibitors in vitro [64]. The rational use of InPeps to identify molecules of therapeutic interest activity have been proven successful [7,8]. To start screening for molecules with intracellular pharmacological activity using a reduced number of InPeps could be advantageous, reducing the costs to identify a prototype molecule of therapeutic interest. Recent technological advances are allowing for the development of ‘drug-like peptides’ that potently and specifically modulate intracellular PPI targets in cell culture and animal models [63,64]. Therefore, peptides have slowly become exciting candidate molecules for improving health and life quality.

## 5. Conclusions

In conclusion, the results presented herein suggest that one of the possible biological functions of InPeps could be the inhibition of PPI. The putative mechanisms underlying the biological significance and therapeutic potential of InPeps deserve further investigations. VFD7-cpp and VFD6-cpp could have potential therapeutic applications to reduce rapamycin and rapalogs undesired side-effects, while VFD6-cpp could have an application to reduce the damage from ischemic cell death as observed after stroke.

## Figures and Tables

**Figure 1 cells-11-00385-f001:**
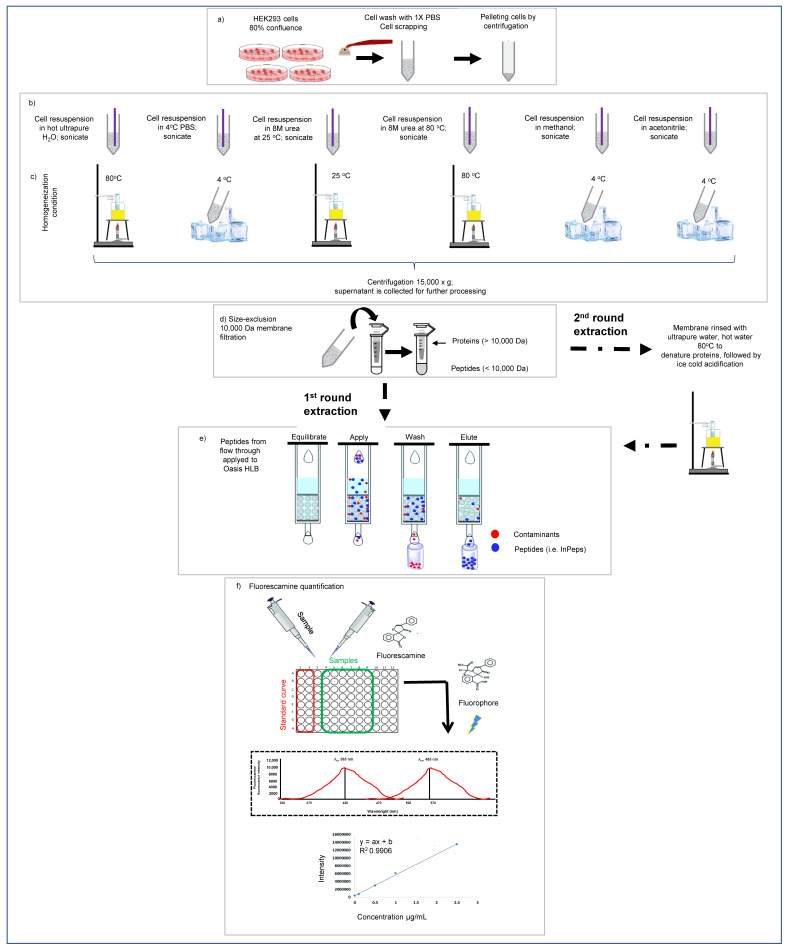
Summarized scheme of InPeps extraction from HEK293 cells. (**a**–**c**) Sub-confluent HEK293 cell cultures were resuspended and homogenized in defined conditions, sonicated, centrifuged, and placed into Amicon^®^ Ultra 4 mL low-binding 10,000 Da size-exclusion filter membranes (Merck-Millipore, Burlington, MA, USA). (**d**) The flowthrough was collected and stored at −20 °C before quantification; InPeps present on this material were considered from “first round” of extraction. The material retained in the 10,000 Da size-exclusion filter membranes was removed by rinsing with ultrapure water, heated at 80 °C for 5 min to denature proteins, acidified with 6N HCl after been ice cold, and stored at −20 °C before quantification; InPeps present on this material were considered “second round” of extraction. (**e**) Stored samples were thawed on ice and applied to Oasis HLB columns for peptide purification as per manufacturer’s instructions. After being dried into a speed vacuum concentrator (Eppendorf, Hamburg, Germany) and resuspended in ultrapure water (60 μL). (**f**) Fluorescamine was used for peptide quantification, as previously described [36,85]. Phosphate buffered saline, pH 7.4 (PBS).

**Figure 2 cells-11-00385-f002:**
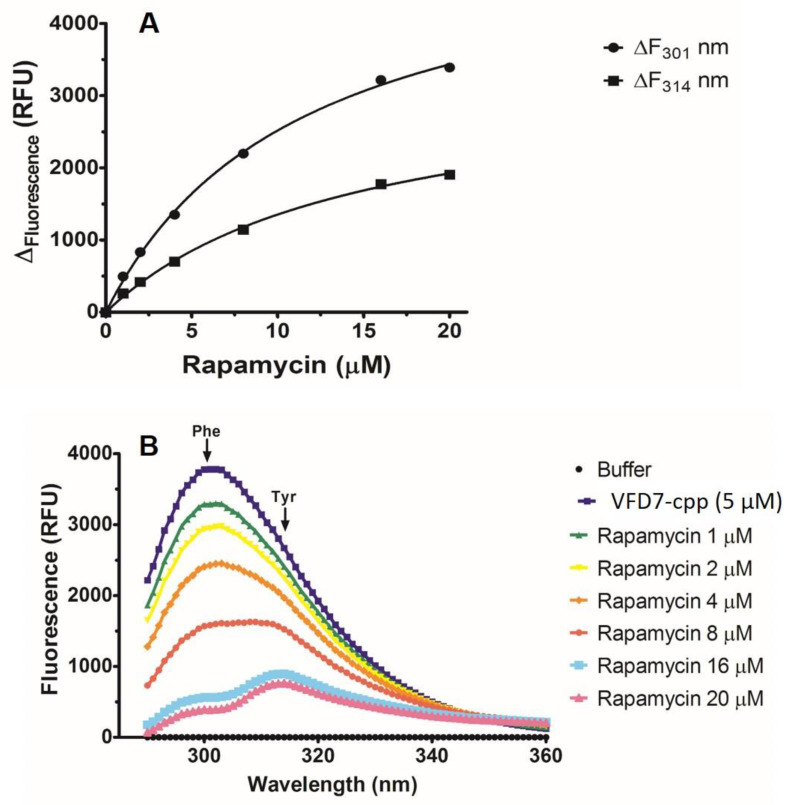
Evaluating the direct molecular interaction between VFD7 and rapamycin. (**A**) The variation (Δ_fluorescence_) of the intrinsic fluorescence emission of the peptide VFD7-cpp (5 µM) at 301 nm (Phe residue) and 314 nm (Tyr residue) in function of the rapamycin concentration (0–20 µM) was monitored in 20 mM Tris-HCl buffer, pH 7.5 at 25 °C. The kinetic analysis shows that rapamycin binds to VFD7-cpp at a single site by a saturable bimolecular reaction with an apparent *K*_d_ of 13 ± 1 µM. (**B**) Changes of VFD7-cpp intrinsic fluorescence emission spectra (290–360 nm) after excitation at λ_ex_ = 275 nm plus/minus rapamycin (0–20 µM). Increasing rapamycin concentration proportionally resulted in a progressive decrease in the fluorescence emission spectra of the peptide. Additional details were presented in Experimental Procedures.

**Figure 3 cells-11-00385-f003:**
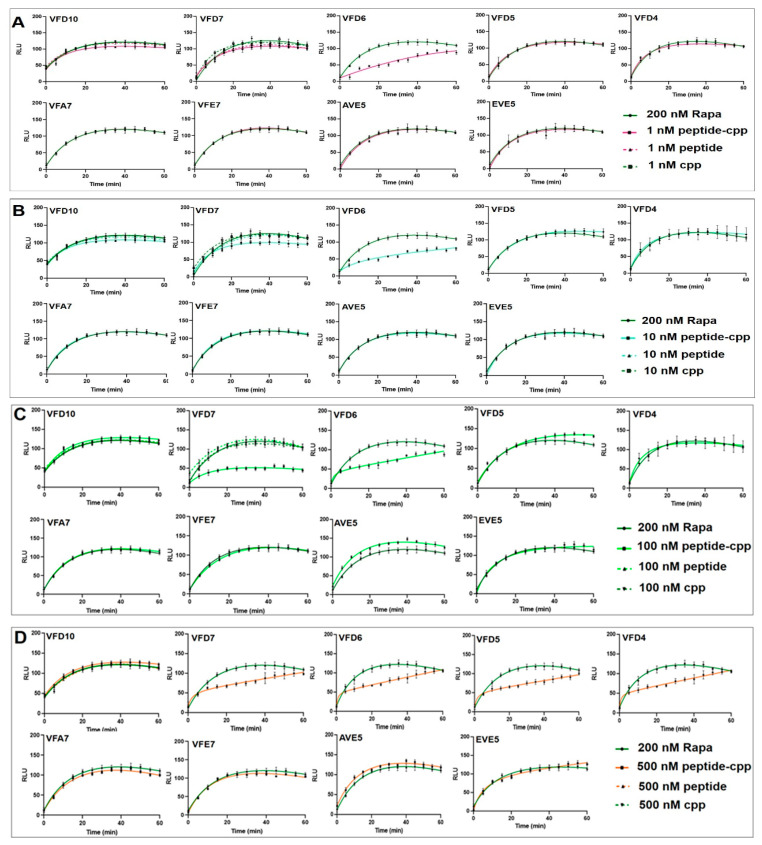
Effect of VFD7-cpp and derived peptides on rapamycin-induced interaction of FKBP–FRB. HEK293 cells co-transfected with NanoBiT^®^ luciferase structural complementation reporter system, as described under Experimental procedures. Note that VFD7-cpp and VFD6-cpp with greatest potencies, and VFD5-cpp and VFD4-cpp with lower potencies, prevented the rapamycin-induced interaction between FKBP–FRB within HEK293 cells. Results are representative of at least five independent determinations conducted in triplicates. Variation among replicates were <10% (error bars). (**A**–**D**); color lanes indicate the experimental condition of each experiment. Relative luminescence units, RLU.

**Figure 4 cells-11-00385-f004:**
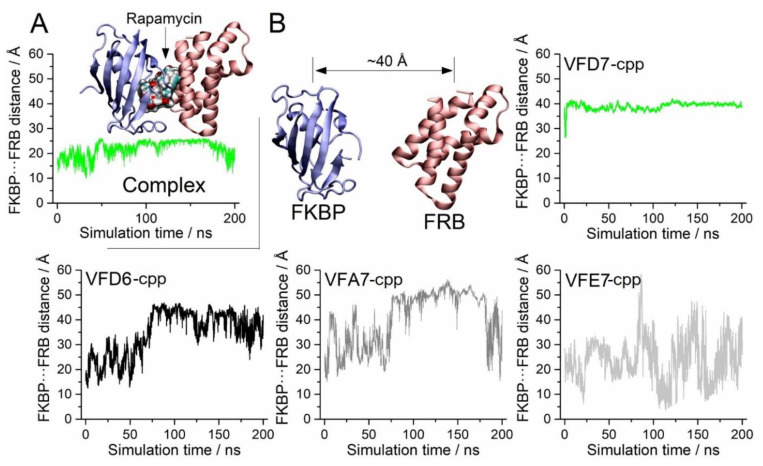
Distances between the FKBP and FRB proteins in the presence of rapamycin and the indicated peptide. (**A**) Considering a pre-stablished FKBP–rapamycin–FRB complex, VFD7-cpp was not able to cause dissociation, keeping the FKBP–FRB distance close to 20 Å (green line) from each protein center of mass. (**B**) VFD7-cpp added previously to the FKBP–rapamycin–FRB complex formation kept at 40 Å, the estimated distance between FKBP and FRB, even in the presence of rapamycin. Peptides VFD6-cpp, VFA7-cpp, and VFE7-cpp were also able to maintain, at least for some time, the 40 Å distance between FKBP and FRB, although not as stabilized as for VFD7-cpp; that was particularly not so evident for VFE7-cpp peptide. All values were considered from the proteins’ centers of mass.

**Figure 5 cells-11-00385-f005:**
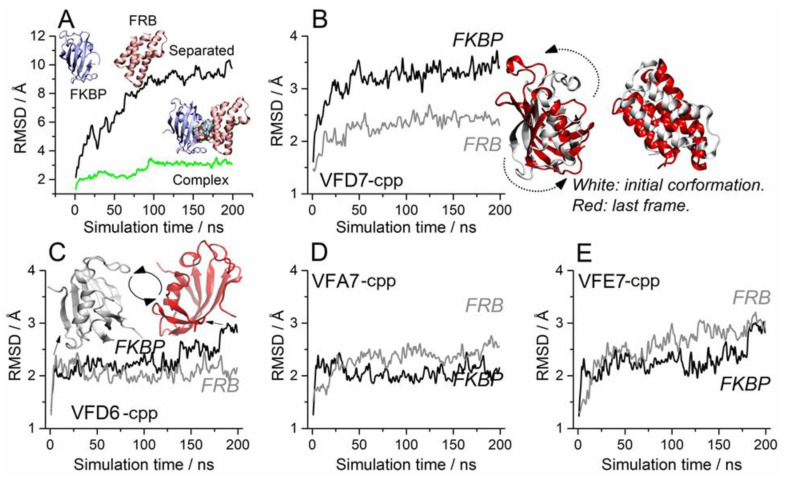
Proteins displacement from RMSD analysis considering the (**A**) the pre-formed FKBP–rapamycin–FRB complex (green line) and the separated chains of FKBP and FRB in a solution containing both rapamycin and VFD7-cpp (black line). The detailed RMSD values for FKBP (black line) and FRB (gray line) were presented as follows: (**B**) VFD7-cpp; (**C**) VFD6-cpp; (**D**) VFA7-cpp; and (**E**) VFE7-cpp. Note that VFD7-cpp seems to alter substantially the initial conformation of FKBP, with comparatively less disturbance promoted on the FRB; a slight FKBP rotation was induced by VFD6-cpp, as seen around 175–200 ns.

**Figure 6 cells-11-00385-f006:**
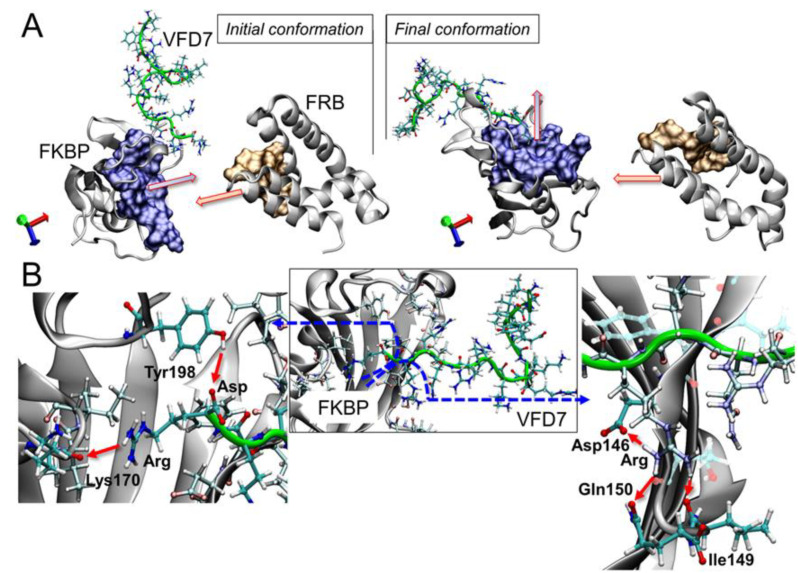
FKBP rotation promoted by the VFD7-cpp peptide. (**A**) shows the misalignment of the FKBP–FRB interfaces; expected contact residues are highlighted on the surface according to the previous study [4], and (**B**) shows the hydrogen interactions which attach the peptide close to FKBP binding interface.

**Figure 7 cells-11-00385-f007:**
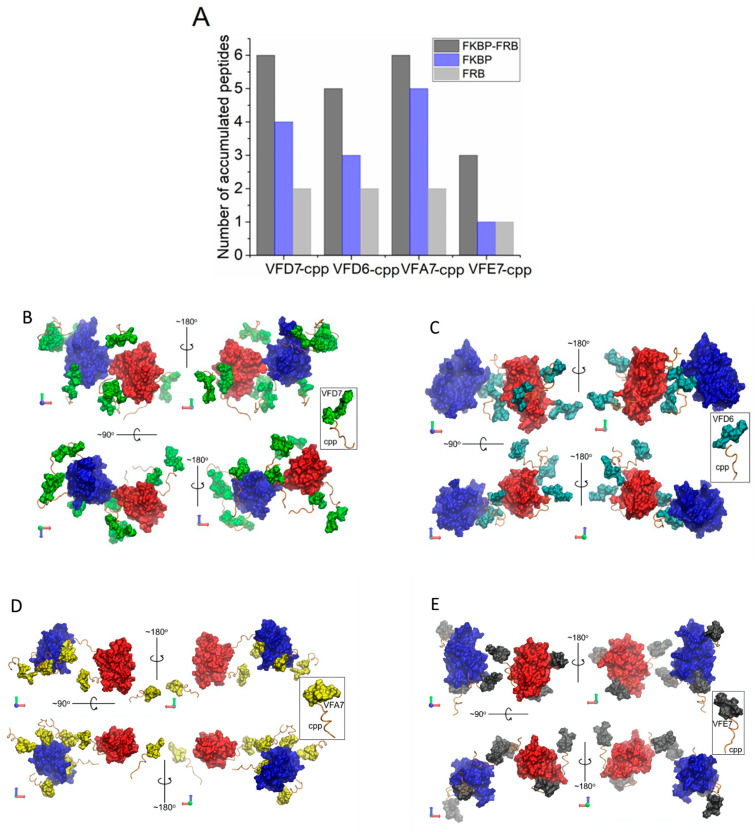
Peptides stabilized on the protein surfaces. (**A**) The average number of peptide molecules close to the FKBP and FRB domains. (**B**–**E**) Structures at the end of the simulation trajectories showing the direct interaction between the peptides and both FKBP and FRB proteins. Highlighted in the small boxes are the peptides’ predicted structures.

**Figure 8 cells-11-00385-f008:**
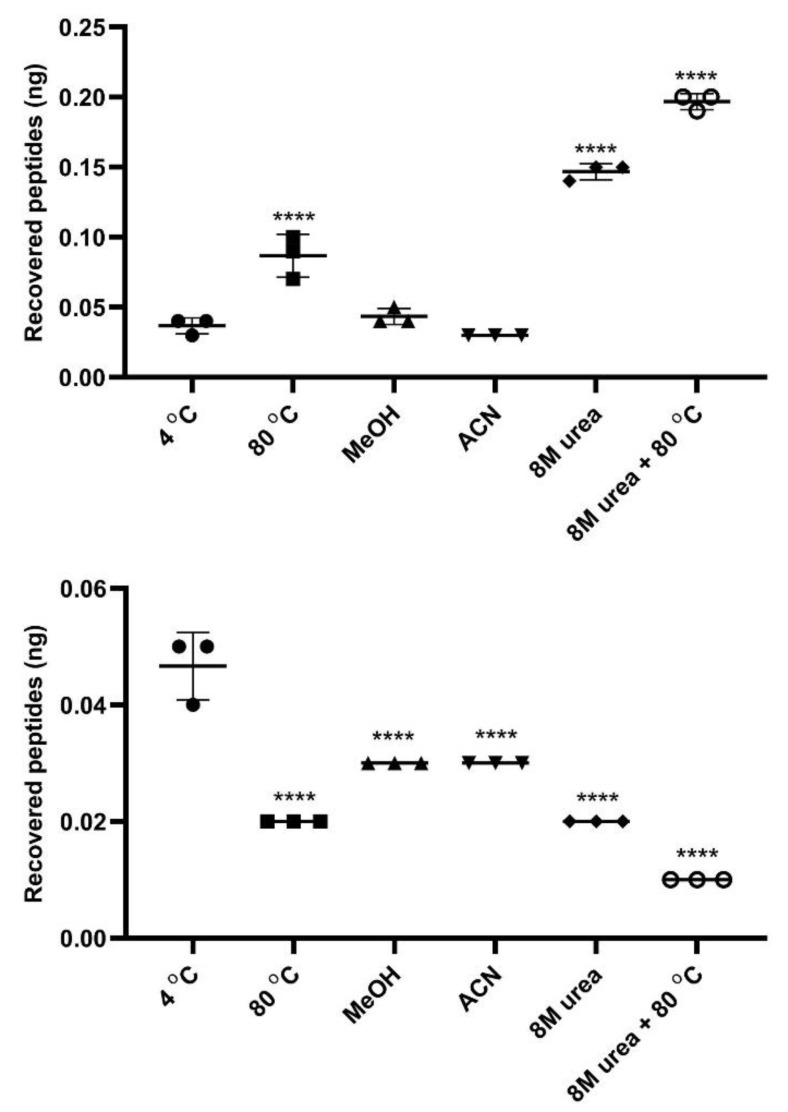
Quantitation of InPeps extracted from HEK293 cells in two independent rounds. Upper panel, first round of InPeps extraction: highest recoveries of InPeps were obtained in the flow-through of 10,000 Da size-exclusion membranes using cells homogenizing conditions that denature proteins and/or nucleic acids structures, such as water at 80 °C (80 °C), 8M urea in water (8 M urea), or 8M urea in water at 80 °C (8 M urea + 80 °C); note that, during this first round, the lowest amounts of InPeps were extracted with PBS at 4 °C (4 °C; smoothest extraction condition), methanol (MeOH), or acetonitrile (ACN) at room temperature. Bottom panel, second round of InPeps extraction: highest recoveries of InPeps were obtained directly from 10,000 Da size-exclusion membranes used to filter cells homogenized with PBS at 4 °C during the first round of extractions, suggesting that extractions using smooth conditions can preserve InPeps associated to macromolecular (i.e., proteins and/or nucleic acids) components of the cells; note that, during the second round, the lowest amounts of InPeps were extracted from 10,000 Da size-exclusion membranes used to filter cells homogenized with either 80 °C, 8 M urea, or 8 M urea + 80 °C in the first round of extractions. Similar numbers of HEK293 cells were homogenized and used for all extractions experiments. InPeps quantification were conducted using fluorescamine. All experiments were performed in triplicate in three independent replicates, which produced similar results. Variations among triplicates were below 5%. MeOH: methanol. ACN: acetonitrile. **** *p* < 0.0001 relative to 4 °C extractions. *n* = 3 independent determinations conducted in triplicates.

**Figure 9 cells-11-00385-f009:**
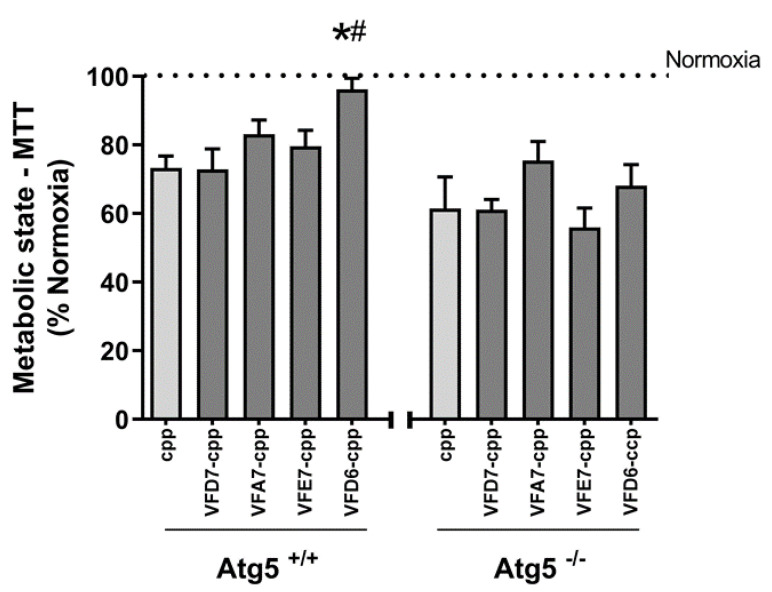
Atg5 wild type (Atg5^+/+^) and Atg5 knockout (Atg5^−/−^) MEF cells were cultured in a 96-well plate in DMEM high-glucose medium supplemented with 10% FBS and 1% penicillin/streptomycin. Cells were washed with PBS and incubated at 37 °C for 3 h in hypoxia buffer, and then re-exposed to oxygen as detailed in Experimental procedures. Treatments with 1μM of each indicated peptide, commenced 30 min before ischemia, and continued during hypoxia and reoxygenation. A booster with the same initial concentration of the peptide was added with the hypoxia buffer incubation and the reoxygenation buffer incubation. Cell viability was measured by MTT assays. Statistics were conducted using ANOVA one-way with post hoc Tukey’s. *, *p* < 0.05 vs. cpp Atg5^+/+^; #, *p* < 0.05 vs. VFD7-cpp Atg5^+/+^; *n* = 3 independent experiments.

**Table 1 cells-11-00385-t001:** Amino acid sequence of peptides investigated herein.

Amino Acid Sequences	Symbol
VFDVELLKLE	VFD10
VFDVELL	VFD7
YGRKKKRRQRRR	cpp
VFDVELLKLEYGRKKKRRQRRR	VFD10-cpp
VFDVELLYGRKKKRRQRRR	VFD7-cpp
VFAVELLYGRKKKRRQRRR	VFA7-cpp
VFEVELLYGRKKKRRQRRR	VFE7-cpp
FDVELLYGRKKKRRQRRR	VFD6-cpp
DVELLYGRKKRRQRRR	VFD5-cpp
VELLYGRKKRRQRRR	VFD4-cpp
EVELLYGRKKKRRQRRR	EVE5-cpp
AVELLYGRKKKRRQRRR	AVE5-cpp

## Data Availability

We hereby state that all data, tables and figures presented in the current manuscript were original, and were never submitted or published in another scientific journal.

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
