# Peer review of "Effect of FKBP12-Derived Intracellular Peptides on Rapamycin-Induced FKBP–FRB Interaction and Autophagy"

_cells, 2022, doi:10.3390/cells11030385_

Round 1
Reviewer 1 Report
The manuscript presented by Parada at al. aims to show that selected peptides produced by proteasome digestion of FKBP12 affect protein-protein interaction between FKBP12 and FRB. Inhibition of this interaction affected autophagy.
Indeed, the role of peptides generated by intracellular proteolytic systems is poorly researched beyond just destine to be recycled. In particular, their participation in PPI seems obvious but not strongly proven.
There are several problems how the data are presented:
- Does VFDVELLKLE peptide participate in native FKPB12 interactions? If so then does it work as a double?
- Do these peptide compete with rapa or they bind in some other place? Results of MD simulations are unclear and a few simple biochemical experiments would help to sort the doubts out. If VFD7 cannot dissociate the complex with RAPA, can RAPA overpower VFD7?
- The tested peptides like VFD7 are not particularly specific and affect function of many proteins so claims of their unique role are not well substantiated
- Other tested peptides are mostly truncated versions of the main subject so it is very difficult with the limited structural inside to judge how it binds to its target.
- Figure 1 would benefit from conversion of the data to a simple set of kinetic data.
- The rationale of using only 200nM RAPA is unclear.
- CCP is just a TAT peptide with its own intracellular functions that also affects proteasome activity. This is problematic in the current setup since it may lead to production of additional peptides derived from FKPB12 and other substrates.
- Tat peptide in solution most likely is not helical even as a CCP – it is not consistent with the published work.
- How stable are these peptides in cellulo? The need for a booster dose suggest that after an hour these peptides are digested to single amino acids.
- Results presented in Fig. 5 suggested that the peptides bind at multiple sites of the complex – again a serious issue of specificity and questioning the results in fig. 4.
- Are these peptides design to protect from RAPA or enhance the effects of RAPA? What would be the therapeutic significance?
- Results of WB were not evaluated by the reviewer in detail but large variances limit they usefulness. Also showing a long line of N.S. data dilutes the message if any.
Author Response
Comments and Suggestions for Authors
The manuscript presented by Parada at al. aims to show that selected peptides produced by proteasome digestion of FKBP12 affect protein-protein interaction between FKBP12 and FRB. Inhibition of this interaction affected autophagy.
Indeed, the role of peptides generated by intracellular proteolytic systems is poorly researched beyond just destine to be recycled. In particular, their participation in PPI seems obvious but not strongly proven.
Answer: We appreciated very much, and would like to thank the reviewer for his/her time and consideration revising our present manuscript. All of your critics, comments and suggestions were acknowledged and very welcome. You may find the present version of the manuscript substantially modified, which was necessary after addressing critics, comments and suggestions made by all reviewers. We hope this revised form of the manuscript is suitable for publication.
Please, find bellow the point-to-point answers to your questions.
There are several problems how the data are presented:
- Does VFDVELLKLE peptide participate in native FKPB12 interactions? If so then does it work as a double?
Answer: Thank you for asking this question. From our data the peptide VFDVELLKLE (VFD10), which was identified in many of our previous mass spectrometry analyses [1; 2; 3], could not inhibit the interaction of FKBP-FRB in our present assays. Therefore, we believe that VFD10 could not participate in vivo of FKBP12 interactions. However, peptide VFDVELL (VFD7) one of the most efficient inhibiting the FKBP-FRB interaction induced by rapamycin could indeed inhibit intracellular FKBP-FRB interaction. As the reviewer mentioned, participation of intracellular peptides generated by the proteasome in PPI “seems obvious”. However, it is technically difficult to directly prove that intracellular peptides generated by proteasome inhibit/regulate PPI, which was the main reason we have used the NanoBit system herein. One of the reasons, proteasome inhibition has many cellular effects by itself, which compromise the possibility of inhibiting the proteasome to evaluate the effects of intracellular peptides. Also, protein point mutation(s) or deletion(s) (for peptides located on the N- or C-terminus) could be used to eliminate the corresponding peptide, but it could compromise the protein function, and further experiments to find out what interactions were altered could not directly evidence the role of the intracellular peptide directly. Therefore, the combination of plasmids expressing proteins that only interact if triggered by a compound (i.e., rapamycin), evaluating the effect of low doses of specific synthetic peptides was a possible model to investigate and suggest that naturally occurring intracellular peptides could also be inhibiting PPI. Of course, we understand that results from the present report are a proof-of-concept to corroborate previous evidences from our group [4; 5], not a definitive demonstration that natural peptides can inhibit PPI. We are unsure if VFD10 or any other evaluated peptide can dimerize to work as a double. We have indeed seen that hemopressin (PVNFKFLSH), a completely different peptide identified by our group in previous reports, can not only dimerize but form fibers in appropriate conditions [6].
Regarding the native interactions and proteolytic stability of the intracellular peptides, we have included in the present revised version of the manuscript additional experiments suggesting the association of intracellular peptides to macromolecular components (i.e. proteins and nucleic acids, per example) of the cells. These new results were presented (Figure 9) and discussed in the current revised version of the manuscript.
- Do these peptides compete with rapa or they bind in some other place? Results of MD simulations are unclear and a few simple biochemical experiments would help to sort the doubts out. If VFD7 cannot dissociate the complex with RAPA, can RAPA overpower VFD7?
Answer: Thank you for asking this question. We have performed biochemical experiments showing that VFD7 at higher concentrations (Phe fluorescence (301 nm) dissociation constant (Kd) was calculated as 11.4 ± 0.9 µM; Tyr fluorescence (314 nm) Kd was calculated as 14.5 ± 1.3 µM) can interact with rapamycin. However, at the concentrations used in present experiments VFD7 and rapamycin interaction, if occurring, was not sufficient to produce the results obtained. In the original version of our manuscript these results were presented as supplementary material. We have now included these data on the main body of the manuscript as Figure 2.
- The tested peptides like VFD7 are not particularly specific and affect function of many proteins so claims of their unique role are not well substantiated
Answer: Thank you for pointing to this question. At the concentration used herein, from 1-1000 nM we expect these peptides to be quite specific. However, we cannot speculate about the possible unspecific effects of these peptides affecting the function of many proteins, because we have not tested that yet. In fact, the present data suggest great specificity of the peptides investigated. During PPI assays only peptides VFD7-cpp and VFD6-cpp, at least in concentrations below 100 nM, were capable to inhibit rapamycin-induced FKBP-FRB interaction, despite the fact that most peptides investigate were rationally designed from VFD7. Moreover, in autophagy tests most peptides could not effectively induce increased cell viability. Only VFD6-cpp was able to protect cells from hypoxia-induced autophagy, only Atg5+/+ but not in Atg5-/- MEF cells. Therefore, if the assumption of the reviewer that these peptides could “affect function of many proteins”, one would expect evident effects for all of the investigate peptides.
- Other tested peptides are mostly truncated versions of the main subject so it is very difficult with the limited structural inside to judge how it binds to its target.
Answer: We have made appropriate corrections to the peptides sequences previous shown on Table 1; we apologize for these previous mistakes that may have confused the reviewer. Peptides rational design included N-terminal deletions of the main VFD10 and VFD7 peptides. However, peptides with individual amino acid substitutions (D for either E or A) were also synthesized (i.e., VFA7-cpp and VFE7-cpp). To address this important issue pointed by the reviewer, we have modified the Experimental procedures session of revised manuscript, including an explanation about the rational used to designing the peptides investigated, as following.
Experimental procedures:
“Moreover, the rationale to design additional peptides was to identify a minimal VFD7/VFD7-cpp-derived sequence retaining the ability to induce similar effects. Thus, VFD7/VFD7-cpp original sequences were successively shortened from amino terminal, maintaining the “hot spot” leucine residues from C-terminus. This rational was based on previous descriptions that similar to protein interactions, peptide binding is mediated by ‘‘hot spot’’ hydrophobic residues that have a high frequency of leucine residues [7; 8]. Additional modifications included a single amino acid substitution, as of aspartic acid (D; side chain pKa 3.9) substituted to either alanine (A; neutral amino acid) or glutamic acid (E; longer side chain, pK 4.3), and also such amino acid substitutions in addition to shorter N-terminal. These VFD7/VFD7-cpp-derived peptide sequences were designed to gain further insight on the structural specificity of the peptides investigated herein, which is usually an important issue of peptides having complex and flexible structures comparable to those of small and rigid molecules.”
Results:
ªThe design of peptides for therapeutic application involves at least a rational step to identify derivatives with increased potencies and smaller number of amino acids; these modifications could reduce the later costs of using peptides in pharmaceutical formulations. Here, the rational of maintaining C-terminus “hot spot” leucine residues [68,69] was partially efficient. Next, the possible effects of VFD7-cpp and derived peptides covalently coupled to cpp on the rapamycin-induced FKBP-FRB interaction were qualitatively evaluated, measuring the RLU generated as a resulting from FKBP-FRB intracellular interaction (Figure 1, A-D). Among peptides evaluated, VFD6-cpp (1-500 nM) followed by VFD7-cpp (10-500 nM) were the most potent inhibiting FKBP-FRB interaction (Figure 2, A-D). Peptides VFD5-cpp and VFD4-cpp although with reduced potencies compared to VFD6-cpp or VFD7-cpp, could inhibit rapamycin-induced FKBP-FRB interaction (Figure 3). However, the presence of these leucine “hot spots” needed to be allied with the presence of the aspartic acid, as the peptides AVE5-cpp and EVE5-cpp, derived from VFD5-cpp, have completely abolished their inhibition of rapamycin-induced FKBP-FRB interaction (Figure 3). Therefore, VFD6-cpp could be suggested as the most potent and smaller VFD7-cpp derivative, capable to inhibit the rapamycin-induced FKBP-FRB interaction.ª
Discussion:
“VFD6-cpp (1-500 nM) and VFD7-cpp (10-500 nM) were the most potent peptides inhibiting rapamycin-induced FKBP-FRB dynamic interaction. The inhibition of PPI at nanomolar concentrations corroborates previous suggestions that natural InPeps have biological significance regulating intracellular protein interactions [4; 5; 9]. At higher doses, VFD5-cpp (500 nM) and VFD4-cpp (500 nM) were also able to inhibit rapamycin-induced FKBP-FRB interaction. Peptides VFD10-cpp, VFE7-cpp, VFA7-cpp, EVE5-cpp and AVE5-cpp were unable to alter rapamycin-induced FKBP-FRB interaction. The VFD7-cpp amino acid substitution from D to A (VFA7-cpp) or to E (VFE7-cpp) completely abolished its inhibition of FKBP-FRB interaction induced by rapamycin. London and colleagues created a unique nonredundant database of high-resolution structures of peptide–protein complexes (termed peptiDB) to investigate the structural basis of free peptide-protein interactions [8]. Free Peptides tend to bind to proteins in a more planar fashion than structured and unstructured proteins do, which may allow peptides to overcome their own configurational entropy loss upon binding [8]. In some cases, protein structure is known to be unstable and could be stabilized only upon binding to the peptides [8]. Similar to protein interactions, peptide binding is mediated by ‘‘hot spot’’ hydrophobic residues that have a high frequency of leucine residues [7; 8]. Peptide domains within protein structures have been shown to exist throughout entire protein structures[10; 11], and could be released intact after proteolytic degradation of proteins by proteasome (Ferro, E.S. and Klitzke, C.F., unpublished data). Although participation of proteasome generated peptides in PPI could seem obvious [9], most previous studies have not considered the fact that free natural InPeps could affect protein interactions within cells. Coincidently, the most frequently identified InPeps generated by the proteasome, which includes the peptide VFD7, have a hydrophobic amino acid, frequently leucine, at the C-terminus [3; 12]. Indeed, substitutions of leucine to alanine totally abolished the cell death activity of peptide WELVVLGKL (pep5) [13]. Herein, the presence of leucine ‘‘hot spot” was considered during the rational design of VFD7-cpp derivatives. Results suggest that preserving the leucine “hot spots” was not enough to preserve peptides derived form VFD7-cpp pharmacological activities as inhibitors of rapamycin-induced FKBP-FRB interaction. The absence of the aspartic acid, as in peptides AVE5-cpp and EVE5-cpp, derived from VFD5-cpp, completely abolished the inhibition of rapamycin-induced FKBP-FRB interaction. Therefore, VFD6-cpp could be suggested as the most potent and smaller VFD7-cpp derivative, capable to inhibit the rapamycin-induced FKBP-FRB interaction.”
- Figure 1 would benefit from conversion of the data to a simple set of kinetic data.
Answer: Thank you for your critical opinion. We choose to make a figure as we understand it could be easier to readers, especially those who are not familiar with kinetic analyses, to understand the inhibitory effects of the peptides in a graphical representation. Moreover, the NanoBiT® luciferase structural complementation reporter system used herein were for qualitative measurement, and definition of kinetic parameters were not on the scope. However, this is an important question that we will certainly address in future experiments. One of the limitations to perform kinetic analyses in living cells that we have to solve before doing these assays, refers to how to measure the kinetics of proteolytic degradation from internalization to the end of the experiments. In other words, these are not trivial kinetics data to analyze.
- The rationale of using only 200nM RAPA is unclear.
Answer: Thank you for your question. We performed a dose-response curve using different concentrations of rapamycin. These data were shown as Supplementary Figure S1 B. Rapamycin at 200 nM nicely stimulate the interaction of FKBP-FRB, giving a luminescent signal that was enough to evaluate possible inhibitions of PPI.
- CCP is just a TAT peptide with its own intracellular functions that also affects proteasome activity. This is problematic in the current setup since it may lead to production of additional peptides derived from FKPB12 and other substrates.
Answer: Thank you for pointing this issue. This was the main reason we were careful making all control experiments in the presence of cpp alone, as well as constructing different peptides with the covalently bound cpp. Therefore, the PPI inhibition observed were mainly due to the peptide evaluated and not due to cpp presence. It is important to mention that the specificity of the cpp used herein have been previously shown [13; 14].
- Tat peptide in solution most likely is not helical even as a CCP – it is not consistent with the published work.
Answer: Thank you for pointing this issue. Peptides in solution generally have flexible structure, except ate the inner amino acids. The peptides predicted structures presented in the current manuscript were obtained with 3 different free and very popular software/servers, such as IUPred2A server[15], DisEMBL server [16], and PEP-FOLD3 tool of the Mobyle framework[17]. These secondary structural predictions were used to perform the molecular dynamics simulations, as described in the manuscript. It was not on the scope of this manuscript to determine the peptides structures in solution, using for example NMR.
- How stable are these peptides in cellulo? The need for a booster dose suggests that after an hour these peptides are digested to single amino acids.
Answer: Thank you for pointing this issue. We have added an additional booster of peptide due to previous experience that the stability of peptides can vary depending on the peptide sequence. We have also included additional experiments to discuss this specific and important question raised by the reviewer (Figure 8). These results are related both to the stability and biological significance of intracellular peptides. Thus, the following paragraph was added to the discussion:
“Hundreds of InPeps have been described from the most diverse sources, including plants [18; 19; 20; 21], yeast [22], fish [23], and mammals [1; 24; 25; 26]. Despite of these strong evidences that InPeps are natural constituents of the cells, the expected short half-life of peptides within cells full of peptidases contests their stability and biological significance [27; 28; 29; 30]. Indeed, most synthetic peptides introduced within cells were shown to have short half-life [4; 27]. For example, after 30 min of internalization into Chinese hamster ovarian cells the concentration of a group of four InPeps was reduced from 1.6% (SSGAHGEEGSARIWKA) to 26% (GSAKVAFSAIRSTNH), whereas a random peptide (VNMVPVGWASFR) used as control was 100% recovered [4]. On the other hand, synthetic fluorescent peptides were shown to compete with endogenous peptides in the nuclear compartment for binding to chromatin [27], which decreases the rate of degradation and suggest association of InPeps to macromolecular components of the cells [4; 27; 31]. This paradox has been limiting a clear understanding about the possible biological significance of InPeps. Herein, extraction of InPeps using different conditions such as heat and 8M urea, well-known to denature proteins, were shown to be more efficient compared to mild extraction with PBS at 4°C. On the other hand, a significant higher recover of peptides occurred from the 10,000 Da size-exclusion filter membranes, if the peptide extraction was conducted at mild conditions using PBS at 4°C. These data suggest that within cells most InPeps were tightly associated to macromolecular components (i.e., proteins and/or nucleic acids), justifying the need of stringent conditions (i.e., heat, 8M urea and acidic pH) to extract these peptides from cells or tissues [12; 22; 32; 33; 34]. These data corroborate previous suggestions that within cells InPeps can scape post-proteasomal proteolysis associating with macromolecular components [27; 31]. In addition, these results suggested that within cells InPeps were also unassociated/free from macromolecular components of the cells (i.e., peptides extracted in PBS at 4 °C). Thus, one possibility to explain the presence of natural InPeps within cells and tissues, in opposition to synthetic peptides introduced into the cells that last for only a short time, could be their binding to macromolecular components of the cells. In addition, to maintain the levels of natural InPeps within cells under strict regulatory control a homeostatic equilibrium must exist [4; 27; 31; 35]. Such mechanisms could help to explain the presence of InPeps within cells and tissues, and the changes observed on InPeps levels in pathological conditions [25; 33; 36; 37; 38; 39; 40], along the different phases of the cell cycle [13], after treatment with interferon gamma [41], induced by thermogenesis in human brown adipose tissue [42], following plant treatment with either phytohormone [18] salycilic acid [19], during diet-induced obesity [43], among others [27; 35; 44; 45; 46; 47]. Together, these data corroborate the main suggestion of the present report (using VFD7-cpp and derivatives as a model), which InPeps can be functional within cells binding and regulating PPI.”
- Results presented in Fig. 5 suggested that the peptides bind at multiple sites of the complex – again a serious issue of specificity and questioning the results in fig. 4.
Answer: Thank you for this question. Molecular dynamics simulations suggested that selected peptides, such as VFD7-cpp, VFD6-cpp, VFAVELLYGRKKKRRQRRR (VFA7-cpp) and VFEVELLYGRKKKRRQRRR (VFA7-cpp), bind to FKBP and to FRB protein surfaces. Although, only VFD7-cpp and VFD6-cpp induced changes on FKBP structure, which could help understanding their mechanism of PPI inhibition.
Moreover, previously affinity chromatograph columns were constructed suggesting that InPeps FE2 and FE3 covalently bound through its NH2 terminus interacted, respectively, with 10 and 6 proteins of related biological function; both FE2 and FE3 were InPeps intracellularly active that modulated isoproterenol and angiotensin II signal transduction[4]. Most of the proteins associated to FE2 and FE3 have functions related to vesicular transport and signal transduction regulation. These data suggesting that binding to multiple proteins could be an intrinsic property of peptides. Once peptides bind to protein motifs/domains of related functions, from the same or distinctive proteins, they could act synergically producing functional consequences of biological/pharmacological relevance. Thus, the flexible and complex structure of peptides compared to small molecules of rigid and very specific binding site, could became an advantage for therapeutic application of peptides as inhibitors of intracellular PPI.
The concentration of the peptides employed herein, from 1-1000 nM, seem in a possible range of their physiological concentration. Only VFD6-cpp was effective protecting cells from hypoxia-induced autophagy. Therefore, within cells it is expected that not only peptides but small molecules as well would bind to multiple sites. The dissociation constant of these interactions is indeed very important to define possible specific pharmacological activity. Our present data suggested that the peptides investigated herein have at least relative pharmacological specificity in the concentrations evaluated. Otherwise, at least in the highest concentrations of 500 nM all peptides should have inhibited the FKBP-FRB protein interaction, and also presented pharmacological activity increasing cell viability in both Atg5+/+ and Atg5-/- at 1000 nM; none of these unspecific expected interactions occurred.
We have added the following paragraphs to the Discussion session:
“MD simulations suggested that VFD7-cpp and VFD6-cpp directly interacted with both FKBP and FRB. Upon VFD7-cpp or VFD6-cpp binding, FKBP structure rotate mispositioning its FRB-interaction domain, which could be related to their inhibition of rapamycin-induced interaction of FKBP-FRB in HEK293 cells. MD simulations also suggested that peptides VFA7-cpp and VFE7-cpp bind to distinctive regions of both FKBP and FRB, whereas experimental data suggested that these predicted interactions could not have disturbed proteins structures enough to be functional. Previously, affinity chromatograph columns were constructed suggesting that InPeps FE2 and FE3 covalently bound through its NH2 terminus interacted, respectively, with 10 and 6 proteins of related biological function; both FE2 and FE3 were InPeps intracellularly active that modulated isoproterenol and angiotensin II signal transduction[4]. Most of the proteins associated to FE2 and FE3 have functions related to vesicular transport and signal transduction regulation. These data suggesting that binding to multiple protein regions/proteins could be an intrinsic property of peptides. Once peptides bind to protein motifs/domains of related functions, from the same or distinctive proteins, they could act synergically producing functional consequences of biological/pharmacological relevance [10; 11]. Thus, the flexible and complex structure of peptides compared to that of small molecules of rigid and more specific binding sites, could became an advantage for therapeutic application of peptides as inhibitors of intracellular PPI.”
“There are limitations of using VFD6-cpp and VFD7-cpp as therapeutic peptides, including the stability needed to reach intact the intracellular milieu after being administrated and absorbed in vivo. Indeed, it is well known that most pharmaceutical industries and laboratories prefer proteolytic stable small molecules of well-defined and rigid chemical structure of improved pharmacokinetics. However, it seem relevant to mention that in the past 5–6 years the U.S. Food and Drug Administration (FDA) have authorized the clinical prescription of more than 15 new peptides or peptide-containing molecules [48; 49]. Moreover, PPI have been challenging targets for traditional drug modalities, while macrocyclic peptides have proven highly effective PPI inhibitors in vitro [50]. The rational use of InPeps to identify molecules of therapeutic interest activity have been proven successful [26; 51]. Start screening for molecules with intracellular pharmacological activity using a reduced number of InPeps could be advantageous, reducing the costs to identify a prototype molecule of therapeutic interest. Recent technological advances are allowing for the development of ‘drug-like peptides’ that potently and specifically modulate intracellular PPI targets in cell culture and animal models [50; 52]. Therefore, peptides have slowly become exciting candidate molecules for improving health and life quality.“
- Are these peptides design to protect from RAPA or enhance the effects of RAPA? What would be the therapeutic significance?
Answer: Thank you for asking this question. One possible therapeutic significance of VFD6-cpp would be to reduce the damage of ischemia, such in stroke, in addition toVFD6-cpp and VFD7-cpp that could reduce the undesired side-effects of rapamycin and rapalogs. We have now mentioned these issues on the manuscript, as following.
“Autophagy plays a crucial role in maintaining cellular homeostasis, and is closely related to the occurrence of variety of human diseases. It is known that autophagy occurs in response to various environmental stresses such as nutrient deficiency, growth factor deficiency, and hypoxia [53]. Many studies have shown that a number of signal transduction pathways are involved in the regulation of autophagy [54; 55]. Some autophagic pathways converge at the mTORC1, which plays a central role in the regulation of autophagy [56; 57; 58; 59]. mTORC1 signaling is switched on by several oncogenic signaling pathways and is accordingly hyperactive in the majority of cancers [60], age-related diseases [61], and recently, COVID-19 [62]. Moreover, lysosome activation induced by mTORC1 inhibition depends on Atg5 during autophagy [63]. Rapamycin induces the formation of the FKBP-rapamycin-FRB complex resulting in inhibition of some of the effects of mTORC1 [64], by restricting substrate access to the mTORC1 serine/threonine-protein kinase site [65]. VFD6-cpp and VFD7-cpp should be stimulating mTORC1 activity, as the most potent peptides investigated herein to inhibit the rapamycin-induced FKBP-FRB interaction. However, only VFD6-cpp was seen to increase cell viability in Atg5+/+, but not on autophagy-deficient Atg5−/− challenged by hypoxia. Inhibition of the mTORC1 with rapamycin is currently the only known pharmacological treatment that increases lifespan in all model organisms studied [66; 67; 68]. However, evident side effects have being experienced by the chronic use of rapamycin, such as metabolic defects that include hyperglycemia, hyperlipidemia, insulin resistance and increased incidence of new-onset type 2 diabetes [69]. Patients taking rapamycin to prevent organ transplant rejection have presented with adverse effects including stomatitis, thrombocytopenia, high serum triglycerides and cholesterol, and impaired wound healing [70]. Therefore, a possible therapeutic application for VFD6-cpp could be the inhibition of rapamycin-induced FKBP-FRB interaction and consequent inhibition of mTORC1. This could be useful to reduce at least some of the undesired side-effects of rapamycin and rapalogs. Moreover, VFD6-cpp could be therapeutic useful to prevent extensive cell death during ischemic situations, such as observed during stroke [71; 72]. However, further investigations remain necessary to evaluate whether in living cells proteasome generated InPeps such as VFD7 could be functional stimulating mTORC1 activity.”
- Results of WB were not evaluated by the reviewer in detail but large variances limit they usefulness. Also showing a long line of N.S. data dilutes the message if any.
Answer: Thank you for suggesting that Western blot experiments were not very useful. The Western blot data were removed from the present version of the manuscript, considering the critical comments received from the reviewers. We understand and intend to persuade future studies to address in details the mechanism of action of VFD6-cpp and VFD7-cpp on autophagy flux.
[1] S. Dasgupta, L.M. Castro, R. Dulman, C. Yang, M. Schmidt, E.S. Ferro, and L.D. Fricker, Proteasome inhibitors alter levels of intracellular peptides in HEK293T and SH-SY5Y cells. PLoS One 9 (2014) e103604.
[2] J.S. Gelman, J. Sironi, I. Berezniuk, S. Dasgupta, L.M. Castro, F.C. Gozzo, E.S. Ferro, and L.D. Fricker, Alterations of the Intracellular Peptidome in Response to the Proteasome Inhibitor Bortezomib. PLoS ONE 8 (2013) e53263-e53263.
[3] L.D. Fricker, J.S. Gelman, L.M. Castro, F.C. Gozzo, and E.S. Ferro, Peptidomic analysis of HEK293T cells: effect of the proteasome inhibitor epoxomicin on intracellular peptides. J Proteome Res 11 (2012) 1981-90.
[4] F.M. Cunha, D.A. Berti, Z.S. Ferreira, C.F. Klitzke, R.P. Markus, and E.S. Ferro, Intracellular peptides as natural regulators of cell signaling. J Biol Chem 283 (2008) 24448-59.
[5] L.C. Russo, A.F. Asega, L.M. Castro, P.D. Negraes, L. Cruz, F.C. Gozzo, H. Ulrich, A.C. Camargo, V. Rioli, and E.S. Ferro, Natural intracellular peptides can modulate the interactions of mouse brain proteins and thimet oligopeptidase with 14-3-3epsilon and calmodulin. Proteomics 12 (2012) 2641-55.
[6] H.M. Dao, S. Parajuli, E. Urena-Benavides, and S. Jo, Self-assembled peptide fibrils with pH-sensitive reversible surface-active properties. Colloid and Interface Science Communications 39 (2020) 100325.
[7] A. Stein, and P. Aloy, Contextual specificity in peptide-mediated protein interactions. PLoS One 3 (2008) e2524.
[8] N. London, D. Movshovitz-Attias, and O. Schueler-Furman, The structural basis of peptide-protein binding strategies. Structure 18 (2010) 188-99.
[9] E.S. Ferro, S. Hyslop, and A.C. Camargo, Intracellullar peptides as putative natural regulators of protein interactions. J Neurochem 91 (2004) 769-77.
[10] T. Pawson, and P. Nash, Protein-protein interactions define specificity in signal transduction. Genes & development 14 (2000) 1027-47.
[11] T. Pawson, and J.D. Scott, Signaling through scaffold, anchoring, and adaptor proteins. Science (New York, N.Y.) 278 (1997) 2075-80.
[12] J.S. Gelman, J. Sironi, L.M. Castro, E.S. Ferro, and L.D. Fricker, Peptidomic analysis of human cell lines. J Proteome Res 10 (2011) 1583-92.
[13] C.B. de Araujo, L.C. Russo, L.M. Castro, F.L. Forti, E.R. do Monte, V. Rioli, F.C. Gozzo, A. Colquhoun, and E.S. Ferro, A novel intracellular peptide derived from g1/s cyclin d2 induces cell death. J Biol Chem 289 (2014) 16711-26.
[14] C.B. de Araujo, L.P. de Lima, S.G. Calderano, F. Silva Damasceno, A.M. Silber, and M.C. Elias, Pep5, a Fragment of Cyclin D2, Shows Antiparasitic Effects in Different Stages of the Trypanosoma cruzi Life Cycle and Blocks Parasite Infectivity. Antimicrob Agents Chemother 63 (2019).
[15] B. Meszaros, G. Erdos, and Z. Dosztanyi, IUPred2A: context-dependent prediction of protein disorder as a function of redox state and protein binding. Nucleic Acids Res 46 (2018) W329-W337.
[16] R. Linding, L.J. Jensen, F. Diella, P. Bork, T.J. Gibson, and R.B. Russell, Protein disorder prediction: implications for structural proteomics. Structure 11 (2003) 1453-9.
[17] B. Neron, H. Menager, C. Maufrais, N. Joly, J. Maupetit, S. Letort, S. Carrere, P. Tuffery, and C. Letondal, Mobyle: a new full web bioinformatics framework. Bioinformatics 25 (2009) 3005-11.
[18] I. Fesenko, R. Azarkina, I. Kirov, A. Kniazev, A. Filippova, E. Grafskaia, V. Lazarev, V. Zgoda, I. Butenko, O. Bukato, I. Lyapina, D. Nazarenko, S. Elansky, A. Mamaeva, V. Ivanov, and V. Govorun, Phytohormone treatment induces generation of cryptic peptides with antimicrobial activity in the Moss Physcomitrella patens. BMC Plant Biol 19 (2019) 9.
[19] A. Filippova, I. Lyapina, I. Kirov, V. Zgoda, A. Belogurov, A. Kudriaeva, V. Ivanov, and I. Fesenko, Salicylic acid influences the protease activity and posttranslation modifications of the secreted peptides in the moss Physcomitrella patens. J Pept Sci 25 (2019) e3138.
[20] I. Fesenko, R. Khazigaleeva, V. Govorun, and V. Ivanov, Analysis of Endogenous Peptide Pools of Physcomitrella patens Moss. Methods Mol Biol 1719 (2018) 395-405.
[21] I.A. Fesenko, G.P. Arapidi, A.Y. Skripnikov, D.G. Alexeev, E.S. Kostryukova, A.I. Manolov, I.A. Altukhov, R.A. Khazigaleeva, A.V. Seredina, S.I. Kovalchuk, R.H. Ziganshin, V.G. Zgoda, S.E. Novikova, T.A. Semashko, D.K. Slizhikova, V.V. Ptushenko, A.Y. Gorbachev, V.M. Govorun, and V.T. Ivanov, Specific pools of endogenous peptides are present in gametophore, protonema, and protoplast cells of the moss Physcomitrella patens. BMC Plant Biol 15 (2015) 87.
[22] S. Dasgupta, C. Yang, L.M. Castro, A.K. Tashima, E.S. Ferro, R.D. Moir, I.M. Willis, and L.D. Fricker, Analysis of the Yeast Peptidome and Comparison with the Human Peptidome. PLoS One 11 (2016) e0163312.
[23] C.M.M. Teixeira, C.N. Correa, L.K. Iwai, E.S. Ferro, and L.M. Castro, Characterization of Intracellular Peptides from Zebrafish (Danio rerio) Brain. Zebrafish 16 (2019) 240-251.
[24] E.S. Ferro, V. Rioli, L.M. Castro, and L.D. Fricker, Intracellular peptides: From discovery to function. EuPA Open Proteomics 3 (2014) 143-151.
[25] G.R. Sakaya, C.A. Parada, R.A. Eichler, V.N. Yamaki, A. Navon, A.S. Heimann, E.G. Figueiredo, and E.S. Ferro, Peptidomic profiling of cerebrospinal fluid from patients with intracranial saccular aneurysms. J Proteomics 240 (2021) 104188.
[26] C.B. de Araujo, A.S. Heimann, R.A. Remer, L.C. Russo, A. Colquhoun, F.L. Forti, and E.S. Ferro, Intracellular Peptides in Cell Biology and Pharmacology. Biomolecules 9 (2019).
[27] E. Reits, A. Griekspoor, J. Neijssen, T. Groothuis, K. Jalink, P. van Veelen, H. Janssen, J. Calafat, J.W. Drijfhout, and J. Neefjes, Peptide diffusion, protection, and degradation in nuclear and cytoplasmic compartments before antigen presentation by MHC class I. Immunity 18 (2003) 97-108.
[28] A. Navon, and A.L. Goldberg, Proteins are unfolded on the surface of the ATPase ring before transport into the proteasome. Mol Cell 8 (2001) 1339-49.
[29] K.L. Rock, I.A. York, and A.L. Goldberg, Post-proteasomal antigen processing for major histocompatibility complex class I presentation. Nat Immunol 5 (2004) 670-7.
[30] A.L. Goldberg, Protein degradation and protection against misfolded or damaged proteins. Nature 426 (2003) 895-9.
[31] P. Paz, N. Brouwenstijn, R. Perry, and N. Shastri, Discrete proteolytic intermediates in the MHC class I antigen processing pathway and MHC I–dependent peptide trimming in the ER. Immunity 11 (1999) 241-251.
[32] J.S. Gelman, S. Dasgupta, I. Berezniuk, and L.D. Fricker, Analysis of peptides secreted from cultured mouse brain tissue. Biochim Biophys Acta 1834 (2013) 2408-17.
[33] I. Berezniuk, J. Sironi, M.B. Callaway, L.M. Castro, I.Y. Hirata, E.S. Ferro, and L.D. Fricker, CCP1/Nna1 functions in protein turnover in mouse brain: Implications for cell death in Purkinje cell degeneration mice. Faseb j 24 (2010) 1813-23.
[34] J.S. Gelman, J. Sironi, L.M. Castro, E.S. Ferro, and L.D. Fricker, Hemopressins and other hemoglobin-derived peptides in mouse brain: comparison between brain, blood, and heart peptidome and regulation in Cpefat/fat mice. J Neurochem 113 (2010) 871-80.
[35] D.A. Berti, C. Morano, L.C. Russo, L.M. Castro, F.M. Cunha, X. Zhang, J. Sironi, C.F. Klitzke, E.S. Ferro, and L.D. Fricker, Analysis of intracellular substrates and products of thimet oligopeptidase in human embryonic kidney 293 cells. J Biol Chem 284 (2009) 14105-16.
[36] L.D. Fricker, Proteasome Inhibitor Drugs. Annu Rev Pharmacol Toxicol 60 (2020) 457-476.
[37] S. Dasgupta, M. Fishman, H. Mahallati, L. Castro, A. Tashima, E. Ferro, and L. Fricker, Reduced Levels of Proteasome Products in a Mouse Striatal Cell Model of Huntington’s Disease. PLOS ONE 10 (2015) e0145333.
[38] I. Berezniuk, J.J. Sironi, J. Wardman, R.C. Pasek, N.F. Berbari, B.K. Yoder, and L.D. Fricker, Quantitative peptidomics of Purkinje cell degeneration mice. PloS one 8 (2013) e60981-e60981.
[39] L.O. Fiametti, C.N. Correa, and L.M. Castro, Peptide Profile of Zebrafish Brain in a 6-OHDA-Induced Parkinson Model. Zebrafish 18 (2021) 55-65.
[40] C.C.C. Café-Mendes, E.S.S. Ferro, A.S.S. Torrão, F. Crunfli, V. Rioli, A. Schmitt, P. Falkai, L.R.R. Britto, C.W.W. Turck, and D. Martins-de-Souza, Peptidomic analysis of the anterior temporal lobe and corpus callosum from schizophrenia patients. Journal of proteomics 151 (2017) 97-105.
[41] E.R. Monte, C. Rossato, R.P. Llanos, L.C. Russo, L.M. de Castro, F.C. Gozzo, C.B. de Araujo, J.P. Peron, O.A. Sant'Anna, E.S. Ferro, and V. Rioli, Interferon-gamma activity is potentiated by an intracellular peptide derived from the human 19S ATPase regulatory subunit 4 of the proteasome. J Proteomics 151 (2017) 74-82.
[42] Y. Li, X. Wang, F. Wang, L. You, P. Xu, Y. Cao, L. Chen, J. Wen, X. Guo, X. Cui, and C. Ji, Identification of intracellular peptides associated with thermogenesis in human brown adipocytes. J Cell Physiol 234 (2019) 7104-7114.
[43] M.C.F. Gewehr, A.A.S. Teixeira, B.A.C. Santos, L.A. Biondo, F.C. Gozzo, A.M. Cordibello, R.A.S. Eichler, P. Reckziegel, R.N.O. Da Silva, N.B. Dos Santos, N.O.S. Camara, A. Castoldi, M.L.M. Barreto-Chaves, C.S. Dale, N. Senger, J. Lima, M.C.L. Seelaender, A.C. Inada, E.H. Akamine, L.M. Castro, A.C. Rodrigues, J.C.R. Neto, and E.S. Ferro, The Relevance of Thimet Oligopeptidase in the Regulation of Energy Metabolism and Diet-Induced Obesity. Biomolecules 10 (2020) E321.
[44] D.M.L.P. Cavalcanti, L.M. Castro, J.C. Rosa Neto, M. Seelaender, R.X. Neves, V. Oliveira, F.L. Forti, L.K. Iwai, F.C. Gozzo, M. Todiras, I. Schadock, C.C. Barros, M. Bader, and E.S. Ferro, Neurolysin knockout mice generation and initial phenotype characterization. The Journal of biological chemistry 289 (2014) 15426-40.
[45] L.M. Castro, D.M. Cavalcanti, C.B. Araujo, V. Rioli, M.Y. Icimoto, F.C. Gozzo, M. Juliano, L. Juliano, V. Oliveira, and E.S. Ferro, Peptidomic analysis of the neurolysin-knockout mouse brain. J Proteomics 111 (2014) 238-48.
[46] N.B.D. Santos, R.D. Franco, R. Camarini, C.D. Munhoz, R.A.S. Eichler, M.C.F. Gewehr, P. Reckziegel, R.P. Llanos, C.S. Dale, V. Silva, V.F. Borges, B.H.F. Lima, F.Q. Cunha, B. Visniauskas, J.R. Chagas, S. Tufik, F.F. Peres, V.C. Abilio, J.C. Florio, L.K. Iwai, V. Rioli, B.C. Presoto, A.O. Guimaraes, J.B. Pesquero, M. Bader, L.M. Castro, and E.S. Ferro, Thimet Oligopeptidase (EC 3.4.24.15) Key Functions Suggested by Knockout Mice Phenotype Characterization. Biomolecules 9 (2019).
[47] D.A. Berti, L.C. Russo, L.M. Castro, L. Cruz, F.C. Gozzo, J.C. Heimann, F.B. Lima, A.C. Oliveira, S. Andreotti, and P.O. Prada, Identification of intracellular peptides in rat adipose tissue: insights into insulin resistance. Proteomics 12 (2012) 2668-2681.
[48] B.G. de la Torre, and F. Albericio, Peptide Therapeutics 2.0, Multidisciplinary Digital Publishing Institute, 2020.
[49] B.G. de la Torre, and F. Albericio, The Pharmaceutical Industry in 2019. An Analysis of FDA Drug Approvals from the Perspective of Molecules. Molecules 25 (2020) 745.
[50] M. Buyanova, and D. Pei, Targeting intracellular protein-protein interactions with macrocyclic peptides. Trends Pharmacol Sci (2021).
[51] P. Reckziegel, W.T. Festuccia, L.R.G. Britto, K.L.L. Jang, C.M. Romao, J.C. Heimann, M.V. Fogaca, N.S. Rodrigues, N.R. Silva, F.S. Guimaraes, R.A.S. Eichler, A. Gupta, I. Gomes, L.A. Devi, A.S. Heimann, and E.S. Ferro, A novel peptide that improves metabolic parameters without adverse central nervous system effects. Sci Rep 7 (2017) 14781.
[52] G.J.B. Philippe, D.J. Craik, and S.T. Henriques, Converting peptides into drugs targeting intracellular protein–protein interactions. Drug Discovery Today 26 (2021) 1521-1531.
[53] B. Levine, and G. Kroemer, Biological Functions of Autophagy Genes: A Disease Perspective. Cell 176 (2019) 11-42.
[54] Y. Wang, and H. Zhang, Regulation of Autophagy by mTOR Signaling Pathway. Adv Exp Med Biol 1206 (2019) 67-83.
[55] K. Wu, A. Seylani, J. Wu, X. Wu, C.K.E. Bleck, and M.N. Sack, BLOC1S1/GCN5L1/BORCS1 is a critical mediator for the initiation of autolysosomal tubulation. Autophagy 17 (2021) 3707-3724.
[56] J. Heitman, N.R. Movva, and M.N. Hall, Targets for cell cycle arrest by the immunosuppressant rapamycin in yeast. Science 253 (1991) 905-909.
[57] Y.-G. Gangloff, M. Mueller, S.G. Dann, P. Svoboda, M. Sticker, J.-F. Spetz, S.H. Um, E.J. Brown, S. Cereghini, and G. Thomas, Disruption of the mouse mTOR gene leads to early postimplantation lethality and prohibits embryonic stem cell development. Molecular and cellular biology 24 (2004) 9508-9516.
[58] M. Laplante, and D.M. Sabatini, mTOR signaling in growth control and disease. Cell 149 (2012) 274-293.
[59] S. Eltschinger, and R. Loewith, TOR complexes and the maintenance of cellular homeostasis. Trends in cell biology 26 (2016) 148-159.
[60] J. Dancey, mTOR signaling and drug development in cancer. Nature reviews Clinical oncology 7 (2010) 209.
[61] H.E. Walters, and L.S. Cox, mTORC inhibitors as broad-spectrum therapeutics for age-related diseases. International journal of molecular sciences 19 (2018) 2325.
[62] E. Bischof, R.C. Siow, A. Zhavoronkov, and M. Kaeberlein, The potential of rapalogs to enhance resilience against SARS-CoV-2 infection and reduce the severity of COVID-19. The Lancet Healthy longevity 2 (2021) e105-e111.
[63] J. Zhou, S.-H. Tan, V. Nicolas, C. Bauvy, N.-D. Yang, J. Zhang, Y. Xue, P. Codogno, and H.-M. Shen, Activation of lysosomal function in the course of autophagy via mTORC1 suppression and autophagosome-lysosome fusion. Cell Research 23 (2013) 508-523.
[64] J. Xie, X. Wang, and C.G. Proud, mTOR inhibitors in cancer therapy. F1000Res 5 (2016).
[65] H. Yang, D.G. Rudge, J.D. Koos, B. Vaidialingam, H.J. Yang, and N.P. Pavletich, mTOR kinase structure, mechanism and regulation. Nature 497 (2013) 217-223.
[66] T. Weichhart, mTOR as Regulator of Lifespan, Aging, and Cellular Senescence: A Mini-Review. Gerontology 64 (2018) 127-134.
[67] S. Horvath, A.T. Lu, H. Cohen, and K. Raj, Rapamycin retards epigenetic ageing of keratinocytes independently of its effects on replicative senescence, proliferation and differentiation. Aging 11 (2019) 3238-3249.
[68] J.E. Wilkinson, L. Burmeister, S.V. Brooks, C.C. Chan, S. Friedline, D.E. Harrison, J.F. Hejtmancik, N. Nadon, R. Strong, and L.K. Wood, Rapamycin slows aging in mice. Aging cell 11 (2012) 675-682.
[69] A.B. Salmon, About-face on the metabolic side effects of rapamycin. Oncotarget 6 (2015) 2585-6.
[70] S.C. Johnson, and M. Kaeberlein, Rapamycin in aging and disease: maximizing efficacy while minimizing side effects. Oncotarget 7 (2016).
[71] S.R. Var, A.V. Shetty, A.W. Grande, W.C. Low, and M.C. Cheeran, Microglia and Macrophages in Neuroprotection, Neurogenesis, and Emerging Therapies for Stroke. Cells 10 (2021).
[72] W.C. Jean, S.R. Spellman, E.S. Nussbaum, and W.C. Low, Reperfusion injury after focal cerebral ischemia: The role inflammation and the the rapeutic horizon. Neurosurgery 43 (1998) 1382-1396.
- Does VFDVELLKLE peptide participate in native FKPB12 interactions? If so then does it work as a double?
Thank you for raising this question. We have used VFDVELLKLE (VFD10), VFDVELL (VFD7), VFDVEL (VFD6), and additional peptides, at nano molar concentrations. It not unreasonable to suppose that intracellular peptides biologically could exist at such namo molar concentrations within cells. Therefore, we believe that our present data reflects what could is occurring naturally, following FKBP12 degradation by the proteasome. Whereas the VFD10 was not able to inhibit rapamycin-induced FKBP-FRB interaction, both VFD7 and VFD6 efficiently inhibited such interaction at nano molar concentrations. We have mentioned in the manuscript that the cellular levels of these peptides were affected by proteasome inhibitions. Therefore, it is possible to suggest that VFD7 and VFD6 are biologically functional inhibiting protein interactions such as FKBP-FRB.
Moreover, it is technically difficult to directly prove that VFD10, VFD7 or VFD6, or any other intracellular peptides, have biological activity regulating protein-protein interactions (PPI). First, to block FKBP12 degradation using proteasome inhibitors will affect the degradation of additional proteins. Second, to inhibit FKBP12 protein synthesis will affect other cellular functions that depend on this protein. Therefore, we have been working on cellular and animal models to provide solid evidences that intracellular peptides are functional. Previously, we have used surface plasmon resonance (SPR) to show that several intracellular peptides could inhibit PPI with distinctive patterns. VFD10 and VFD7 were two of the most efficient peptides to block PPI investigated by SPR. However, within the cellular environment the present report is the first one to directly show intracellular peptides regulating PPI, at nano molar concentration and with great structural specificity.
- Do these peptide compete with rapa or they bind in some other place? Results of MD simulations are unclear and a few simple biochemical experiments would help to sort the doubts out. If VFD7 cannot dissociate the complex with RAPA, can RAPA overpower VFD7?
Thank you for raining such question. We have investigated this question and results were presented as Supplemental Figure 7 (results session, first paragraph). VFD7 at concentrations from 11-14 micro Molar can interact with rapamycin. We performed all experiments at concentrations at least 200 fold bellow this concentration. Therefore, VFD7 do not complexed with rapamycin to produce the observed results from blocking FKBP-FRB interaction.
- The tested peptides like VFD7 are not particularly specific and affect function of many proteins so claims of their unique role are not well substantiated
Thank you for this critical comments. The Gaussia luciferase structural complementation reporter system NanoBiT® (Promega, WI, USA) used herein is commercially available. There are previous reports showing that only the interaction of FKBP-FRB could reestablish the Gaussia luciferase functional activity. Therefore, even thought that VFD7 and other intracellular peptides could have multiple protein targets (possibly with distinctive constants of dissociations), which is quite expectable because peptides have complex structures compared to small molecules, in the present report the effects of these peptides were specifically inhibiting the interaction of FKBP-FRB induced by rapamycin.
- Other tested peptides are mostly truncated versions of the main subject so it is very difficult with the limited structural inside to judge how it binds to its target.
Thank you for raising this question. Indeed, not only truncations but also amino acids substitutions were made. Please, note that Val was substituted by both Glu and Ala, and also Asp was substituted by Ala and Glu. Therefore, the complete set of peptides investigated herein presented a strong structure-function relationship concluding that VFD7 and VFD6 were functional among other similar sequences.
- Figure 1 would benefit from conversion of the data to a simple set of kinetic data.
We thank the reviewer for these suggestions. Indeed, to make a table with the kinetics and dissociation constants in the prsence and absence of the peptides would help. However, we would like to keep the figure as it is, because it illustrates quite rapidly what the manuscript major point, which is that intracellular peptides can modulate/inhibit protein interactions.
- The rationale of using only 200nM RAPA is unclear.
Supplemental Figure S1, B shows dose response curves of rapamycin. The concentration of 200 nM was sufficient to reach almost the maximum luminescence of the Gaussia luciferase system.
- CCP is just a TAT peptide with its own intracellular functions that also affects proteasome activity. This is problematic in the current setup since it may lead to production of additional peptides derived from FKPB12 and other substrates.
In addition to VFD7 and VFD6, we have used several peptides containing the cpp sequence, and cpp itself was also used as a control peptide in all assays. None of the other peptides affected the FKBP-FRB interaction similarly to VFD6 and VFD7. Therefore, it is quite difficult to suggest that by inhibiting proteasome activity cpp was performing the function of inhibiting FKBP-FRB interaction induced by rapamycin.
- Tat peptide in solution most likely is not helical even as a CCP – it is not consistent with the published work.
- How stable are these peptides in cellulo? The need for a booster dose suggest that after an hour these peptides are digested to single amino acids.
In previous reports we have seen that some peptides last longer, and other peptides last not too long. In the present report we have not investigated the stability of the peptides.
- Results presented in Fig. 5 suggested that the peptides bind at multiple sites of the complex – again a serious issue of specificity and questioning the results in fig. 4.
As mentioned before, peptides have complex structures and could bind to multiple targets.
- Are these peptides design to protect from RAPA or enhance the effects of RAPA? What would be the therapeutic significance?
- Results of WB were not evaluated by the reviewer in detail but large variances limit they usefulness. Also showing a long line of N.S. data dilutes the message if any.

Reviewer 2 Report
General remarks and question:
- To properly familiarize the reader with scope and subject of the manuscript, Authors should enrich the ‘Introduction’ section by:
- Information about cell penetrating peptides (together with premises to select the YGRKKKRRQRRR peptide for this particular study);
- mTORC1 regulation (by rapamycin and studied peptides) from the perspective of later evaluated autophagy induction;
- Information about working principle of utilized Gaussia luciferase structural complementation reporter system NanoBiT®.
- Authors should state a clear objective in each subsection of Experimental procedures (especially subsections 2.1, 2.3 and 2.4)
- No detailed peptide synthesis protocol is provided in reference [24] (line 99). Authors should clearly describe the peptide synthesis methodology including description of:
- used resin
- deprotection (reagents and time);
- acylation (reagents and time);
- cleavage and side-chain deprotection (reagents and time);
- HPLC purification process (equipment, column, elution profile);
- HPLC analysis (evaluation of purity; equipment, column, elution profile);
- Counter-ion (if exchanged, if not, authors should state that it is TFA salt);
- Chromatograms presenting peptide purity should be included in supplementary file.
- Subsection 5 Structural models for the VFD-cpp peptide and variants presents the results of applied test instead of it description.
- The experiment described in Results section (lines 255 – 275) should be described in Experimental procedures section (also clear objective should be included)
- Authors should clearly distinguish InPeps from their CCP derivatives. For example, VFD7 should be regarded as different molecule than VFD7-cpp. Nevertheless, across the manuscript Authors refer to both molecules as VFD7.
- Subsection Western blot assays (Experimental procedures) is not in line with Possible intracellular biological significance of VFD7 and related peptides (Results). Specifically, experimental setup should clearly describe all tested combinations of compounds and premises for evaluating co-treatment, while Results section should describe all observed properties (even ‘no changes’).
Additional question to Authors:
Why no significant difference was observed between rapamycin (positive control) and control or chloroquine (both negative controls) in any of molecular markers of autophagy (Figure 7)? If no statistically significant difference is observed between positive and negative controls what is the reason to evaluate this test as credible?
- Authors should elaborate in Discussion section on:
- Applicability of rapamycin-induced FKBP-FRB interaction in pharmacological (molecule design) and biological field.
- Potential mode of action of studied peptides (with special emphasis on co-treatment with chloroquine)
- Potential reason for the fact that peptides increased cell viability after hypoxia-reoxygenation in vitro
Minor remarks:
- Rapamycin is generally recognized as immunosuppressant drug instead of antifungal antibiotic (line 61).
- DMEM – no explanation of this abbreviation is provided in the text (line 108).
- Plus/minus – with/without is more applicable to this case (line 129).
- CYS 2085 was used instead of what? (line 165).
- Figure 6 – Atg5 has no superscript (line 405).
- (…) to investigated (…) – should be corrected (line 409).
Author Response
Comments and Suggestions for Authors General remarks and question:
Authors Comments:
Answer: We appreciated very much, and would like to thank the reviewer for his/her time and consideration revising our present manuscript. All of your critics, comments and suggestions were acknowledged and very welcome. You may find the present version of the manuscript substantially modified, which was necessary after addressing critics, comments and suggestions made by all reviewers. We hope this revised form of the manuscript is suitable for publication.
Please, find bellow the point-to-point answers to your questions.
To properly familiarize the reader with scope and subject of the manuscript, Authors should enrich the ‘Introduction’ section by:
- Information about cell penetrating peptides (together with premises to select the YGRKKKRRQRRR peptide for this particular study);
Answer: We have included the following information about cpp on the Introduction:
“Cell penetrating peptides (cpp) are relatively short peptides, 4–40 aa, frequently used to carry associated molecules into the cells mainly by endocytosis [54]. The covalent coupling of peptides to cpp constitutes a useful alternative to CHAPS transient membrane permeabilization, allowing easier cell internalization to investigate possible intracellular functions of peptides [55-57].”
- mTORC1 regulation (by rapamycin and studied peptides) from the perspective of later evaluated autophagy induction;
Answer: We would like to thank the reviewer for raising this point. We have made an extensive edition on the Introduction to address this important issue. Please, find below the edition added to the Introduction.
“Autophagy original definition was the delivery of cytoplasmic cargo to the lysosome for degradation. There are at least three distinct forms of autophagy (i.e., chaperone-mediated autophagy, microautophagy and macroautophagy), which differ in terms of mode of cargo delivery to the lysosome [38]. Macroautophagy is the major catabolic mechanism used by eukaryotic cells to maintain nutrient homeostasis and organellar quality control. It is mediated by a set of evolutionarily conserved genes, the autophagy-related genes (Atg). With a few exceptions, all Atg are required for the efficient formation of sealed autophagosomes that proceed to fuse with lysosomes [38]. The mechanistic target of rapamycin complex 1 (mTORC1) is a member of the phosphatidylinositol 3-kinase-related kinase family of protein kinases [39]. Under nutrient-rich conditions mTORC1 promotes cell growth by stimulating synthesis of proteins, lipids and nucleotides, and by inhibiting cellular catabolism through repression of the autophagic pathway [39-41]. Consensus is increasing that mTORC1 activation occurs at the surface of the lysosomal membrane in response to changes in amino acid sufficiency, which is transduced via the Rag family of small GTPases to mediate the translocation of mTORC1 from the cytoplasm to the surface of the lysosome, where mTORC1 is activated by GTP-binding protein Rheb [42]. mTORC1 integrates various stimuli and signaling networks, while autophagy constitutes an important avenue for nutrient supply providing amino acids to restore mTORC1 activity. mTORC1 localization to the lysosome in autophagy is subject to intricate control by metabolic regulatory networks [42-44]. The activity of mTORC1 toward many substrates is acutely sensitive to rapamycin, but mTORC1 also possess rapamycin-resistant activity toward certain substrates [45-47].
The protein complex formation between 12-kDa FK506-binding protein (FKBP12) and FKBP12-rapamycin binding domain (FRB) results in mTORC1 inhibition, by restricting substrate access to the mTORC1 serine/threonine-protein kinase site [48,49]. Rapamycin shows many beneficial effects in mice, while in humans rapamycin and rapamycin derivatives (rapalogs) have been used primarily as immunosuppressants following organ transplantation, and in the treatment of several specific types of cancer, including renal cell carcinoma, pancreatic neuroendocrine tumors, and HER2-negative breast cancer [47]. Inhibition of the mTORC1 pathways by rapamycin and rapalogs have also been suggested to extend lifespan [50]. Rapamycin has also clinical application as a medication to prevent organ transplant rejection, and was approved by the FDA to preserve renal allografts under the generic name sirolimus [42,51]. Serious side effects of rapamycin and rapalogs observed in humans include an increased incidence of viral and fungal infections including pneumonia, chronic edema, painful oral aphthous ulceration, and hair loss [52,53]. Metabolic effects of long-term rapamycin treatment have also been observed, including decreased insulin sensitivity, glucose intolerance, and an increased risk of new-onset diabetes [47,53].”
- Information about working principle of utilized Gaussia luciferase structural complementation reporter system NanoBiT®.
Answer: We would like to thank the reviewer for raising this point. We have made an extensive edition both on Introduction and Discussion to address this important issue. Please, find below the edition added, respectively, to the Introduction and Discussion.
Introduction:
“Here, commercially available (NanoLuc Binary Technology, NanoBiT®; Promega, WI, USA) structural complementation reporter system, constructed from deep sea shrimp Oplophorus gracilirostris small luciferase subunit [58,59], was used to investigate the effect of VFD7/VFD7-cpp and derivatives on the formation of the FKBP-rapamycin-FRB complex that within cells results in mTORC1 inhibition [48]. The potency of rapamycin to induce FKBP and FRB protein dimerization was previously shown to be minimally affected by FKBP-FRB intrinsic affinity (range >100,000) [58]. The cpp (YGRKKRRQRRR) used herein was previously shown to transport its cargo mainly into the cytosol [60-62]. VFD7-cpp (10-500 nM) and FDVELL (VFD6) covalently coupled to cpp (VFD6-cpp; 1-500 nM), were the most potent peptides inhibiting rapamycin-induced FKBP-FRB interaction measured using the NanoBiT® technology. Molecular dynamics (MD) simulations suggested a possible mechanism for VFD7-cpp and VFD6-cpp to inhibit rapamycin-induced FKBP-FRB, which involves FKBP structural change and/or direct binding of these peptides to the FKBP-FRB protein interaction domains. VFD6-cpp (1µM) increased cell viability in Atg5+/+, but not autophagy-deficient Atg5−/− mouse embryonic fibroblasts. Moreover, extraction of InPeps using different conditions contributed with a possible mechanism for their intracellular proteolytic stability, and also corroborate with the suggestions that InPeps can be natural modulators of protein interactions within cells [18,34,63,64].”
Discussion:
“PPI investigated herein used the NanoBiT® technology, designed for qualitative investigation of protein interaction dynamics under relevant physiological conditions. Benefiting from the small size and bright luminescence of deep-sea shrimp Oplophorus gracilirostris small luciferase subunit, it provides PPI detection at low intracellular concentrations with minimal steric interference on appended target proteins [58,59]. The intrinsic affinity and association constants determined for NanoBiT® were described to be outside of the ranges typical for protein interactions [58,59]. For the specific interaction of FKBP-FRB, it showed second-order behavior consistent with a dynamic binary system, while at higher concentrations favoring full association of the dimeric complex [58,59]. The luminescent response observed was rapid and reversible without evident lag, indicating that NanoBiT® can react nearly instantaneously to changing intracellular conditions [58,59]. Furthermore, dynamic processes can be monitored for more than one hour due to the stable signal duration of the luminescent reaction [58].”
- Authors should state a clear objective in each subsection of Experimental procedures (especially subsections 2.1, 2.3 and 2.4).
Answer: We would like to thank the reviewer for raising this point. We have made these editions accordingly through the Experimental procedure’s session.
- No detailed peptide synthesis protocol is provided in reference [24] (line 99). Authors should clearly describe the peptide synthesis methodology including description of:
- used resin
- deprotection (reagents and time);
- acylation (reagents and time);
- cleavage and side-chain deprotection (reagents and time);
- HPLC purification process (equipment, column, elution profile);
- HPLC analysis (evaluation of purity; equipment, column, elution profile);
- Counter-ion (if exchanged, if not, authors should state that it is TFA salt);
- Chromatograms presenting peptide purity should be included in supplementary file.
Answer: We apologize for the missing information. We have now provided the information requested by the reviewer regarding peptide synthesis and included the chromatograms from cpp-bound peptides quality control. Chromatograms for VFD10 and VFD7 without cpp-bound were similar (not shown). We have made appropriate corrections to the peptides sequences previous shown on Table 1; we apologize for these previous mistakes that may have confused the reviewer.
- Subsection 5, Structural models for the VFD-cpp peptide and variants presents the results of applied test instead of it description.
Answer: We would like to thank the reviewer for raising this point. We have change it accordingly.
- The experiment described in Results section (lines 255 – 275) should be described in Experimental procedures section (also clear objective should be included)
Answer: We would like to thank the reviewer for raising this point. We have change it accordingly.
- Authors should clearly distinguish InPeps from their CCP derivatives. For example, VFD7 should be regarded as different molecule than VFD7-cpp. Nevertheless, across the manuscript Authors refer to both molecules as VFD7.
Answer: We would like to thank the reviewer for raising this point. VFD10, VFD7 and shorter peptides without the cpp covalently bound were used only as control peptides. However, the peptides that shown evident pharmacological activity were VFD7-cpp and VFD6-cpp. We have change it accordingly in the text.
- Subsection Western blot assays (Experimental procedures) is not in line with Possible intracellular biological significance of VFD7 and related peptides (Results). Specifically, experimental setup should clearly describe all tested combinations of compounds and premises for evaluating co-treatment, while Results section should describe all observed properties (even ‘no changes’).
Answer: Answer: We thank you for pointing this irregularity. The Western blot data were removed from the present version of the manuscript, considering the critical comments received from the other two reviewers. We understand and intend to persuade future studies to address in details the mechanism of action of VFD6-cpp and VFD7-cpp on autophagy flux. We agree with reviewers’ comments that only minor effects were observed, and these effects could not be clearly explained to be related to the activation of mTORC1 caused by VFD6-cpp and VFD7-cpp. We have changed the manuscript accordingly.
Additional question to Authors:
Why no significant difference was observed between rapamycin (positive control) and control or chloroquine (both negative controls) in any of molecular markers of autophagy (Figure 7)? If no statistically significant difference is observed between positive and negative controls what is the reason to evaluate this test as credible?
Answer: We would like to thank the reviewer for raising this point. As mentioned above, these Western blotting experiments were removed from the present version of the manuscript; many additional experiments should be conducted in order to correlate the results obtained with the function/targets of VFD6-cpp and VFD7-cpp. However, we should point that we could indeed observe the expected effects of chloroquine used as a control on the levels of LC3I/II.
Authors should elaborate in Discussion section on:
- Applicability of rapamycin-induced FKBP-FRB interaction in pharmacological (molecule design) and biological field.
Answer: We thank you for point this issue for discussion. We have now added the following paragraphs to the Discussion session:
“Moreover, lysosome activation induced by mTORC1 inhibition depends on Atg5 during autophagy [96]. Rapamycin induces the formation of the FKBP-rapamycin-FRB complex resulting in inhibition of some of the effects of mTORC1 [41], by restricting substrate access to the mTORC1 serine/threonine-protein kinase site [48]. VFD6-cpp and VFD7-cpp should be stimulating mTORC1 activity, as the most potent peptides investigated herein to inhibit the rapamycin-induced FKBP-FRB interaction. However, only VFD6-cpp was seen to increase cell viability in Atg5+/+, but not on autophagy-deficient Atg5−/− challenged by hypoxia. Inhibition of the mTORC1 with rapamycin is currently the only known pharmacological treatment that increases lifespan in all model organisms studied [129-131]. However, evident side effects have being experienced by the chronic use of rapamycin, such as metabolic defects that include hyperglycemia, hyperlipidemia, insulin resistance and increased incidence of new-onset type 2 diabetes [101]. Patients taking rapamycin to prevent organ transplant rejection have presented with adverse effects including stomatitis, thrombocytopenia, high serum triglycerides and cholesterol, and impaired wound healing [100]. Therefore, a possible therapeutic application for VFD6-cpp could be the inhibition of rapamycin-induced FKBP-FRB interaction and consequent inhibition of mTORC1. This could be useful to reduce at least some of the undesired side-effects of rapamycin and rapalogs. Moreover, VFD6-cpp could be therapeutic useful to prevent extensive cell death during ischemic situations, such as observed during stroke [98,132]. However, further investigations remain necessary to evaluate whether in living cells proteasome generated InPeps such as VFD7 could be functional stimulating mTORC1 activity.
There are limitations of using VFD6-cpp and VFD7-cpp as therapeutic peptides, including the stability needed to reach intact the intracellular milieu after being administrated and absorbed in vivo. Indeed, it is well known that most pharmaceutical industries and laboratories prefer proteolytic stable small molecules of well-defined and rigid chemical structure of improved pharmacokinetics. However, it seem relevant to mention that in the past 5–6 years the U.S. Food and Drug Administration (FDA) have authorized the clinical prescription of more than 15 new peptides or peptide-containing molecules [133,134]. Moreover, PPI have been challenging targets for traditional drug modalities, while macrocyclic peptides have proven highly effective PPI inhibitors in vitro [64]. The rational use of InPeps to identify molecules of therapeutic interest activity have been proven successful [7,8]. Start screening for molecules with intracellular pharmacological activity using a reduced number of InPeps could be advantageous, reducing the costs to identify a prototype molecule of therapeutic interest. Recent technological advances are allowing for the development of ‘drug-like peptides’ that potently and specifically modulate intracellular PPI targets in cell culture and animal models [63,64]. Therefore, peptides have slowly become exciting candidate molecules for improving health and life quality.”
- Potential mode of action of studied peptides (with special emphasis on co-treatment with chloroquine)
Answer: Thank you for suggesting this point. Because in the new revised version of the manuscript chloroquine use was not mentioned, we made the option to comment about the possible therapeutic use of VFD6-cpp and VFD7-cpp to reduce the undesired side effects caused by long-term treatment with rapamycin. The two paragraphs already mentioned right above were used to discuss the important issues raised by the reviewer.
- Potential reason for the fact that peptides increased cell viability after hypoxia-reoxygenation in vitro
Answer: We thank you for these suggestions. We have now added the following paragraph to the discussion session: “VFD6-cpp was the unique peptide evaluated herein seen to increase cell viability in Atg5+/+, but not on autophagy-deficient Atg5−/− challenged by hypoxia. Inhibition of the mTORC1 with rapamycin is currently the only known pharmacological treatment that increases lifespan in all model organisms studied [27-29]. However, evident side effects have being experienced by the chronic use of rapamycin, such as metabolic defects that include hyperglycemia, hyperlipidemia, insulin resistance and increased incidence of new-onset type 2 diabetes [19]. Patients taking rapamycin to prevent organ transplant rejection have presented with adverse effects including stomatitis, thrombocytopenia, high serum triglycerides and cholesterol, and impaired wound healing [20]. Therefore, inhibiting rapamycin-induced FKBP-FRB interaction that should activates mTORC1 activity, both VFD6-cpp and VFD7-cpp could be useful to reduce at least some of the undesired side-effects of rapamycin and rapalogs. Moreover, VFD6-cpp could be therapeutic useful to prevent extensive cell death during ischemic situations, such as observed during stroke [21,22]. Further investigations remain necessary to evaluate whether in living cells proteasome generated InPeps such as VFD7 could be functional stimulating mTORC1 activity.”
Minor remarks:
Rapamycin is generally recognized as immunosuppressant drug instead of antifungal antibiotic (line 61).
Answer: We would like to thank the reviewer for raising this point. We have change it accordingly.
DMEM – no explanation of this abbreviation is provided in the text (line 108).
Answer: We would like to thank the reviewer for raising this point. We have change it accordingly.
Plus/minus – with/without is more applicable to this case (line 129).
Answer: We would like to thank the reviewer for raising this point. We have change it accordingly.
CYS 2085 was used instead of what? (line 165).
Answer: We have change it accordingly, as following.
“The crystallographic structure of the FKBP-rapamycin-FRB complex was obtained from the Protein Data Bank (PDB) database, with a resolution of 1.67 Å (PDB code 5GPG) [73]. Subsequently, to study the impact of the peptides on the FKBP-rapamycin-FRB complex, we kept (i) both proteins complexed with rapamycin (using the x y z coordinates of 5GPG structure), in the presence of forty peptide molecules (VFD7-cpp) in aqueous solution (24,650 water molecules) in a cubic box of side 98 Å.”
Figure 6 – Atg5 has no superscript (line 405).
Answer: We would like to thank the reviewer for raising this point. We have change it accordingly.
(…) to investigated (…) – should be corrected (line 409).
Answer: We would like to thank the reviewer for raising this point. We have change it accordingly.
- Langel, Ü. Cell-Penetrating Peptides and Transportan. Pharmaceutics 2021, 13, 987.
- Wender, P.A.; Mitchell, D.J.; Pattabiraman, K.; Pelkey, E.T.; Steinman, L.; Rothbard, J.B. The design, synthesis, and evaluation of molecules that enable or enhance cellular uptake: peptoid molecular transporters. Proceedings of the National Academy of Sciences 2000, 97, 13003-13008.
- Duchardt, F.; Ruttekolk, I.R.; Verdurmen, W.P.; Lortat-Jacob, H.; Bürck, J.; Hufnagel, H.; Fischer, R.; Van den Heuvel, M.; Löwik, D.W.; Vuister, G.W. A cell-penetrating peptide derived from human lactoferrin with conformation-dependent uptake efficiency. Journal of Biological Chemistry 2009, 284, 36099-36108.
- Shoari, A.; Tooyserkani, R.; Tahmasebi, M.; Löwik, D. Delivery of Various Cargos into Cancer Cells and Tissues via Cell-Penetrating Peptides: A Review of the Last Decade. Pharmaceutics 2021, 13, doi:10.3390/pharmaceutics13091391.
- Saxton, R.A.; Sabatini, D.M. mTOR Signaling in Growth, Metabolism, and Disease. Cell 2017, 168, 960-976, doi:10.1016/j.cell.2017.02.004.
- Rabanal-Ruiz, Y.; Otten, E.G.; Korolchuk, V.I. mTORC1 as the main gateway to autophagy. Essays Biochem 2017, 61, 565-584, doi:10.1042/EBC20170027.
- Xie, J.; Wang, X.; Proud, C.G. mTOR inhibitors in cancer therapy. F1000Res 2016, 5, doi:10.12688/f1000research.9207.1.
- Yu, L.; McPhee, C.K.; Zheng, L.; Mardones, G.A.; Rong, Y.; Peng, J.; Mi, N.; Zhao, Y.; Liu, Z.; Wan, F. Termination of autophagy and reformation of lysosomes regulated by mTOR. Nature 2010, 465, 942-946.
- Kim, Y.C.; Guan, K.-L. mTOR: a pharmacologic target for autophagy regulation. The Journal of clinical investigation 2015, 125, 25-32, doi:10.1172/JCI73939.
- Kang, S.A.; Pacold, M.E.; Cervantes, C.L.; Lim, D.; Lou, H.J.; Ottina, K.; Gray, N.S.; Turk, B.E.; Yaffe, M.B.; Sabatini, D.M. mTORC1 phosphorylation sites encode their sensitivity to starvation and rapamycin. Science 2013, 341.
- Thoreen, C.C.; Chantranupong, L.; Keys, H.R.; Wang, T.; Gray, N.S.; Sabatini, D.M. A unifying model for mTORC1-mediated regulation of mRNA translation. Nature 2012, 485, 109-113.
- Lamming, D.W. Inhibition of the Mechanistic Target of Rapamycin (mTOR)-Rapamycin and Beyond. Cold Spring Harb Perspect Med 2016, 6, doi:10.1101/cshperspect.a025924.
- Yang, H.; Rudge, D.G.; Koos, J.D.; Vaidialingam, B.; Yang, H.J.; Pavletich, N.P. mTOR kinase structure, mechanism and regulation. Nature 2013, 497, 217-223.
- Takahara, T.; Amemiya, Y.; Sugiyama, R.; Maki, M.; Shibata, H. Amino acid-dependent control of mTORC1 signaling: a variety of regulatory modes. Journal of Biomedical Science 2020, 27, 87, doi:10.1186/s12929-020-00679-2.
- Mahé, E.; Morelon, E.; Lechaton, S.; Sang, K.-H.L.Q.; Mansouri, R.; Ducasse, M.-F.; Mamzer-Bruneel, M.-F.; de Prost, Y.; Kreis, H.; Bodemer, C. Cutaneous Adverse Events in Renal Transplant Recipients Receiving Sirolimus-Based Therapy1. Transplantation 2005, 79, 476-482.
- Li, J.; Kim, S.G.; Blenis, J. Rapamycin: one drug, many effects. Cell metabolism 2014, 19, 373-379.
- Dixon, A.S.; Schwinn, M.K.; Hall, M.P.; Zimmerman, K.; Otto, P.; Lubben, T.H.; Butler, B.L.; Binkowski, B.F.; Machleidt, T.; Kirkland, T.A., et al. NanoLuc Complementation Reporter Optimized for Accurate Measurement of Protein Interactions in Cells. ACS Chemical Biology 2016, 11, 400-408, doi:10.1021/acschembio.5b00753.
- Hall, M.P.; Unch, J.; Binkowski, B.F.; Valley, M.P.; Butler, B.L.; Wood, M.G.; Otto, P.; Zimmerman, K.; Vidugiris, G.; Machleidt, T., et al. Engineered luciferase reporter from a deep sea shrimp utilizing a novel imidazopyrazinone substrate. ACS Chem Biol 2012, 7, 1848-1857, doi:10.1021/cb3002478.
- Salmon, A.B. About-face on the metabolic side effects of rapamycin. Oncotarget 2015, 6, 2585-2586, doi:10.18632/oncotarget.3354.
- Johnson, S.C.; Kaeberlein, M. Rapamycin in aging and disease: maximizing efficacy while minimizing side effects. Oncotarget 2016, 7.
- Var, S.R.; Shetty, A.V.; Grande, A.W.; Low, W.C.; Cheeran, M.C. Microglia and Macrophages in Neuroprotection, Neurogenesis, and Emerging Therapies for Stroke. Cells 2021, 10, doi:10.3390/cells10123555.
- Jean, W.C.; Spellman, S.R.; Nussbaum, E.S.; Low, W.C. Reperfusion injury after focal cerebral ischemia: The role inflammation and the the rapeutic horizon. Neurosurgery 1998, 43, 1382-1396.
- de la Torre, B.G.; Albericio, F. Peptide Therapeutics 2.0. Multidisciplinary Digital Publishing Institute: 2020.
- de la Torre, B.G.; Albericio, F. The Pharmaceutical Industry in 2019. An Analysis of FDA Drug Approvals from the Perspective of Molecules. Molecules 2020, 25, 745.
- Buyanova, M.; Pei, D. Targeting intracellular protein-protein interactions with macrocyclic peptides. Trends Pharmacol Sci 2021, 10.1016/j.tips.2021.11.008, doi:10.1016/j.tips.2021.11.008.
- Philippe, G.J.B.; Craik, D.J.; Henriques, S.T. Converting peptides into drugs targeting intracellular protein–protein interactions. Drug Discovery Today 2021, 26, 1521-1531, doi:https://doi.org/10.1016/j.drudis.2021.01.022.
- Weichhart, T. mTOR as Regulator of Lifespan, Aging, and Cellular Senescence: A Mini-Review. Gerontology 2018, 64, 127-134, doi:10.1159/000484629.
- Horvath, S.; Lu, A.T.; Cohen, H.; Raj, K. Rapamycin retards epigenetic ageing of keratinocytes independently of its effects on replicative senescence, proliferation and differentiation. Aging 2019, 11, 3238-3249, doi:10.18632/aging.101976.
- Wilkinson, J.E.; Burmeister, L.; Brooks, S.V.; Chan, C.C.; Friedline, S.; Harrison, D.E.; Hejtmancik, J.F.; Nadon, N.; Strong, R.; Wood, L.K. Rapamycin slows aging in mice. Aging cell 2012, 11, 675-682.

Reviewer 3 Report
The authors describe the biological effect and possible molecular mechanism of the action of intracellular FKBP12-derived peptide that is generated by the ubiquitin-proteasome system.
Minor comments.
In the Discussion section, the authors need to discuss the possible mechanism(s) VFD6-cpp peptide may increase survival of ATG5 +/+ but not ATG5 -/- MEF cells, including possible pathways involved in the observed phenomena.
Lines 409-410 In the sentence "Western blots to investigated the levels of LC3BI, LC3BII and SQSTM1/p62 were conducted..." the word "to investigated" should be put in the correct form.
Author Response

(The authors gave the same response as above.)

Round 2
Reviewer 2 Report
1)
“Peptide synthesis resin was Rink amide resin for amide C-terminals or pre-loaded Wang resin for free acid C-terminals.” (lines 570 – 571)
Authors should indicate in Table 1 which petides are amides
2)
“20% piperidine in dimethylformamide (DMF) were used for washing” (Lines 580-581)
Authors should check if it is not a mistake – using such mixture to washing procedure could lead to synthesis failure (the same mixture is used for deprotection).
3)
“These experiments were intended to assess how much natural occurring InPeps could be associated to macromolecular components of the cells (i.e., proteins and/or nucleic acids), which could contribute both to their proteolytic stability and biological function.” (Lines 771 – 774)
It is not 100% clear what Authors mean in this sentence. Please rephrase it.
4)
“These data suggests that VFD7-CPP and VFD6 could be preventing the FKBP- FRB protein-protein interaction complex induced by rapamycin, altering the structure of FKBP-FRB needed for appropriated orientation in protein-protein complex formation; no important changes in such orientation were observed for peptides VFA7 and VFE7. A functional consequence of such structural disturbance could be the absence of the heterodimer FKBP-FRB formation, leading to mTORC1 signaling modulation” (Lines 1250 -1255)
Authors should notice that -CPP suffix was not added to VFD6, VFA7 and VFE7, although the -cpp analogs of those peptides were studied in this experiment
5)
Figure 8
Two graphs should be described accordingly.
Author Response
Authors Comments:
We would like to reinforce our appreciation and to thank you again for your time and consideration revising our present manuscript. All your suggestions were very welcome. We hope this revised form of the manuscript is now suitable for publication.
Please, find bellow the point-to-point answers to your questions.
1) “Peptide synthesis resin was Rink amide resin for amide C-terminals or pre-loaded Wang resin for free acid C-terminals.” (lines 570 – 571). Authors should indicate in Table 1 which petides are amides
Answer: we thank you for your correction. All peptides synthesized herein were free acid C-terminus. We have removed the description of the Rink amide resin for amide C-terminals.
2) “20% piperidine in dimethylformamide (DMF) were used for washing” (Lines 580-581). Authors should check if it is not a mistake – using such mixture to washing procedure could lead to synthesis failure (the same mixture is used for deprotection).
Answer: we thank you for your correction, it was indeed a mistake. The text was appropriately modified and now reads:
“Peptide synthesis used standard Fmoc solid-phase syntheses [66,67], with activating reagent 2-(1H-benzotriazol-1-yl)-1,1,3,3,-tetramethyluronium hexafluorophosphate (HBTU) on an ABI 431 synthesizer. Fmoc deprotection was done on the instrument with 20% piperidine for about 10 minutes.”
3) “These experiments were intended to assess how much natural occurring InPeps could be associated to macromolecular components of the cells (i.e., proteins and/or nucleic acids), which could contribute both to their proteolytic stability and biological function.” (Lines 771 – 774). It is not 100% clear what Authors mean in this sentence. Please rephrase it.
Answer: we acknowledge the reviewer for point this confused sentence. We have modified the text that now reads:
“2.7 InPeps extraction and quantification
These experiments were intended to assess the fraction of natural occurring InPeps associated to macromolecular components of the cells (i.e., proteins and/or nucleic acids), which could contribute both to their proteolytic stability and biological function; thus, it is conceivable that peptides associated to proteins can regulate PPI and, at the same time, be protected from proteases and peptidases that usually hydrolyze peptides/proteins as single-chain.”
4) “These data suggests that VFD7-CPP and VFD6 could be preventing the FKBP- FRB protein-protein interaction complex induced by rapamycin, altering the structure of FKBP-FRB needed for appropriated orientation in protein-protein complex formation; no important changes in such orientation were observed for peptides VFA7 and VFE7. A functional consequence of such structural disturbance could be the absence of the heterodimer FKBP-FRB formation, leading to mTORC1 signaling modulation” (Lines 1250 -1255). Authors should notice that -CPP suffix was not added to VFD6, VFA7 and VFE7, although the -cpp analogs of those peptides were studied in this experiment
Answer: we acknowledge the reviewer for point this mistakes. We have now added the -cpp suffix to peptides VFD6, VFa7 and VFE7 accordingly.
5) Figure 8. Two graphs should be described accordingly.
Answer: we acknowledge the reviewer for pointing this weak o four manuscript. We have improved the figure 8 legend accordingly to your comments. The description of the panels were made using Upper panel and Bottom panel with the intention to keep the figure clean from using letters (i.e., A and B); we hopw it is ok.
“Figure 8. Quantitation of InPeps extracted from HEK293 cells in two independent rounds. Upper panel, first round of InPeps extraction: highest recoveries of InPeps were obtained in the flow-through of 10.000 Da size-exclusion membranes using cells homogenizing conditions that denature proteins and/or nucleic acids structures, such as water at 80°C (80°C), 8M urea in water (8M urea) or 8M urea in water at 80°C (8M urea + 80°C); note that, during this first round the lowest amounts of InPeps were extracted with PBS at 4°C (4°C; smoothest extraction condition), methanol (MeOH) or acetonitrile (ACN) at room temperature. Bottom panel, second round of InPeps extraction: highest recoveries of InPeps were obtained directly from 10.000 Da size-exclusion membranes used to filter cells homogenized with PBS at 4°C during the first round of extractions, suggesting that extractions using smooth conditions can preserve InPeps associated to macromolecular (i.e., proteins and/or nucleic acids) components of the cells; note that, during the second round the lowest amounts of InPeps were extracted from 10.000 Da size-exclusion membranes used to filter cells homogenized with either 80°C, 8M urea or 8M urea + 80°C in the first round of extractions. Similar number of HEK293 cells were homogenized and used for all extractions experiments. InPeps quantification were conducted using fluorescamine. All experiments were performed in triplicates in three independent replicates, which produced similar results. Variation among triplicates were below 5%. MeOH: methanol. ACN: acetonitrile. ****P<0.0001 relative to 4°C extractions. n = 3 independent determinations conducted in triplicates.”